# Sediment supply from lateral moraines to a debris covered glacier in the Himalaya

Teun van Woerkom[1], Jakob F. Steiner[1], Philip D.A. Kraaijenbrink[1], Evan S. Miles[2], and Walter W. Immerzeel[1]

[1]Utrecht University, Department of Physical Geography, PO Box 80115, Utrecht, The Netherlands
[2]School of Geography, University of Leeds, Leeds, UK

**Correspondence:** Teun van Woerkom (t.a.a.vanwoerkom@uu.nl)

**Abstract.** Debris-covered glaciers in the Himalaya play an important role in the high-altitude water cycle. The thickness of the debris layer is a key control of the melt rate of those glaciers, yet little is known about the relative importance of the three potential sources of debris supply: the rockwalls, the glacier bed and the lateral moraines. In this study we hypothesize that mass movement from the lateral moraines is a significant debris supply to debris-covered glaciers, in particular when the glacier is disconnected from the rockwall due to downwasting. To test this hypothesis eight high-resolution and highly accurate digital elevation models from the lateral moraines of the debris-covered Lirung Glacier in Nepal are used. These are created using Structure from Motion (SfM), based on images captured using an unmanned aerial vehicle between May 2013 and April 2018. The analysis shows that mass transport results in an elevation change on the lateral moraines with an average rate of -0.31±0.26 m yr$^{-1}$ during this period, partly related to sub-moraine ice melt. There is a higher elevation change rate observed in the monsoon (-0.39±0.74 m yr$^{-1}$) than in the dry season (-0.23±0.68 m yr$^{-1}$). The lower debris aprons of the lateral moraines decrease in elevation at a faster rate during both seasons, probably due to the melt of ice below. The surface lowering rates of the upper gullied moraine, with no ice core below, translate into an annual increase in debris thickness of 0.08 m yr$^{-1}$ along a narrow margin of the glacier surface, with an observed absolute thickness of approximately 1 m, reducing melt rates of underlying glacier ice. Further research should focus on how large this negative feedback is in controlling melt and how debris is redistributed on the glacier surface.

## 1 Introduction

Glaciers cover approximately 110000 km$^2$ in High Mountain Asia (HMA) and as such constitute an important water storage of the region (Pfeffer et al., 2014; Immerzeel et al., 2010). The glaciers are likely to melt rapidly in the future with projections ranging from a total mass loss of 36% to 64% in the coming century depending on the climate scenario (Kraaijenbrink et al., 2017). While the whole HMA has experienced an overall glacier mass and area loss in recent decades, the changes have been found to be variable in space (Bolch et al., 2012; Kääb et al., 2012; Brun et al., 2017). Apart from differences in climate, the presence of debris with variable thickness on the glacier tongues plays an important role to explain this spatial heterogeneity (Scherler et al., 2011; Gardelle et al., 2012; Kääb et al., 2012).

Although the debris-covered area only constitutes 11% of the total glacier area, 30% of the ice mass below the equilibrium line altitude (ELA) is covered in debris (Kraaijenbrink et al., 2017). While a thin debris cover increases melt because it decreases albedo, debris thickness above a critical thickness inhibits melt (Östrem, 1959). The surface of a debris-covered glacier is often characterized by ice cliffs and supraglacial ponds, which result in high local melt rates (Immerzeel et al., 2014b; Steiner et al., 2015; Thompson et al., 2016; Miles et al., 2017a). Knowledge about actual debris thickness is limited to few field observations (Nicholson and Benn, 2006; Ragettli et al., 2015; McCarthy et al., 2017) and attempts to derive it from thermal bands of satellite imagery (Mihalcea et al., 2008; Förster et al., 2012; Rounce and McKinney, 2014; Schauwecker et al., 2015; Rounce et al., 2018). Several studies suggest that glaciers with prolonged periods of negative mass balance are associated with an increase in debris cover (Deline, 2005; Mihalcea et al., 2006; Stokes et al., 2007; Shukla et al., 2009; Kirkbride and Deline, 2013a; Gibson et al., 2017a). Glaciers that have a neutral or positive mass balance, such as in the Karakoram in Pakistan, do not show any positive trends in debris-covered area (Herreid et al., 2015).

Not only is the debris-covered area variable, but the thickness also changes over time. A recent study on the Baltoro glacier in Pakistan suggested that debris thickness changes at rates of multiple centimeters per year with a maximum of 30 cm yr$^{-1}$ near the snout (Gibson et al., 2017a).

The source of supraglacial debris has been studied for decades (Reheis, 1975; Boulton, 1978). Three potential sources of supraglacial debris can be distinguished, which can be divided into direct and indirect sources. The direct sources are basal erosion and bordering rockwalls. The indirect source is the lateral moraine, from which temporarily deposited debris is remobilized and transported (Figure 1A). The relative contribution of these sources is variable, but generally depends on the amount and intensity of precipitation, glacier size, rockwall extent and erodibility of the bedrock and sediments (Benn and Ballantyne, 1994; Benn and Owen, 2002). In HMA, basal erosion (Figure 1A-a) is likely only of limited relative importance, as the erosion processes on the steep rockwalls and lateral moraines are highly active (Benn and Ballantyne, 1994; Benn and Owen, 2002). Bordering rockwalls have been observed to transport debris to the glacier via rockfall, rock avalanches and landslides as primary transporting processes (Figure 1A-b). These periodic events can be triggered by processes such as extreme rainfall or seismic events, and are known to increase due to glacier melt and downwasting of the surface resulting in debuttressing and exposure of unsupported strata (Soldati et al., 2004; Deline et al., 2014).

Generally, long-term rockwall erosion rates are in the order of 1 mm yr$^{-1}$ (Heimsath and McGlynn, 2008). However, the actual mass deposited on the glacier surface is more difficult to determine and varies from glacier to glacier, depending on its exposure to rockwalls and geologic conditions. The reworking of these deposits on the glacier surface, as a result of local hillslope geomorphology, surface melt, glacier flow and englacial processes, may lead to small-scale variations in debris cover as found in the Himalaya. In contrast to transport from the rockwalls, which deposits material on specific locations, remobilization of lateral moraine debris can result in a much more spatially uniform debris supply to the glacier (Figure 1A-c). Lateral moraines typically have a gullied upper part, where debris flows are the main transport process (Curry et al., 2006) (Figure 1B-c1). Most of these flows are released due to rainfall saturation, but the melt of buried ice cores can contribute as well (Ballantyne, 2013). The oversteepened upper parts may also be susceptible to rockfall (Lukas et al., 2012) (Figure 1B-c2). Eroded gully material is deposited near the glacier margin as coalescing debris cones, which together with rockfall and slumping forms the lower

debris apron of the moraine. The material on these debris aprons is reworked by frost action, small landslides, debris flows, snow avalanches and solifluction (Figure 1B-c3).

During downwasting, the glacier surface becomes lower than the moraine crest, making lateral moraine slopes susceptible to mass transport processes. As more and more of the moraine slopes become exposed, a larger amount of moraine-debris can be transported to the glacier surface and their relative contribution to the sediment budget will increase. This is particularly the case for stagnating glacier tongues, where the distal slope of the lateral moraine has the opposite aspect of the adjacent valley slopes (Figure 3). Due to glacier retreat and downwasting, the valley wall is disconnected from the actual tongue. Material deposited below the ELA will furthermore remain at the glacier surface and add to the supraglacial debris cover. Estimated rates of vertical lowering on the lateral moraines range from 49 to 151 mm yr$^{-1}$ for the European Alps (Curry et al., 2006) and 3 to 169 mm yr$^{-1}$ for Norwegian field sites (Curry et al., 2006). On glaciers in the study area (Langtang Valley in Nepal) a supply rate of $0.4 - 31$ mm yr$^{-1}$ (Watanabe et al., 1998) was previously found. However these values are averages over a much longer time span (<550 years versus <247 years for studies in Norway and <79 for the Alps, (Curry et al., 2006)) or include source areas beyond the moraines (Watanabe et al., 1998). These rates are nonetheless all much higher than rockwall erosion rates. All eroded material is deposited on debris cones (often on top of the glacier ice), which are intensely reworked throughout the study period.

Previous studies only reported glacier-averaged erosion rates, although it is likely that the rates are spatially variable. We hypothesize that transport of remobilized debris from lateral moraines can explain the thick and continuous debris cover of tongues on downwasting debris-covered glaciers, that have been disconnected from headwalls, in High Mountain Asia. To this end, we use multi-annual, high-resolution orthomosaics and digital elevation models (DEM) acquired using an unmanned aerial vehicle to quantify surface lowering rates of the lateral moraines of a debris-covered glacier in Nepal. We attempt to explain the spatio-temporal variability and sediment transport processes using the terrain morphology and prevailing seasonal climates. Finally, we assess how important erosion from lateral moraines is in the formation of Himalayan debris-covered glaciers.

## 2 Study area

The research was conducted on Lirung Glacier (28.23 N, 85.56 E), which is located in the Langtang Valley in the Nepalese Himalaya (Figure 2B). The glacier has a southern aspect, steep rockwalls near the mountain crest and an almost flat terminus. The lower part of the glacier is covered entirely in debris (Figure 4) and has been subject to strong downwasting in recent decades (Immerzeel et al., 2014a; Nuimura et al., 2017). The study area is located between 3900 and 4400 m a.m.s.l. (Figure 2C, orange lines). In this section debris from the valley rockwall cannot reach the glacier surface due to the opposite aspect of the distal moraine slopes (Figure 3). The climate in the region is dominated by the monsoon, with a wet season between June and September in which 70% of the annual precipitation falls. During the dry season between November and May, considerably less precipitation reaches the area, and falls mostly as snow (Immerzeel et al., 2014b).

## 3 Data and methods

### 3.1 Field data

Our observations of the lateral moraines of Lirung Glacier span 2013 to 2018, consisting of images captured during multiple UAV flights with an optical camera. A Structure from Motion (SfM) workflow (Lucieer et al., 2014) is used to derive three-dimensional (3-D) point clouds, which is georeferenced by marking measured GCPs and tie points. These operations finally result in co-registered orthomosaics (0.1 m resolution) and DEMs (0.2 m resolution, for further details regarding measurements, processing and data quality see Kraaijenbrink et al. (2016)). Details about the mapped part of the glacier for each UAV campaign are provided in Table 1. The mutually overlapping area of all datasets covers 1.5 km$^2$ of the glacier tongue including the moraines. The accuracy of all generated DEMs is tested by comparing the differences in DEM and GCPs.

Precipitation data was available from the meteorological station at Kyanjing Village (28.21 N, 85.57 E) about 1.3 km south of the glacier in 2013. As data gaps are present between 2015 and 2018 for the Kyanjing station, the meteorological station in Langshisha (28.20 N, 85.67 E) about 13 km south-east of the glacier was used for the remainder of the period. Both rainfall intensities and the cumulative rainfall between any two UAV time slices were analyzed.

### 3.2 Deriving change in elevation

The DEMs were used to calculate vertical elevation differences between time steps. Two preprocessing steps were taken. First, vegetated areas were selected using a maximum likelihood supervised classification applied to the orthomosaic and masked out of the DEMs, to remove noise related to vegetation growth and decay. Second, as the off-glacier and off-moraine terrain did not show signs of sediment transport during the investigated time span, these areas are assumed to be stable. Therefore the DEMs were corrected for elevation changes in off-moraine and off-glacier terrain. To not bias the result towards possible errors in the DEM, we removed the outliers outside the 10-90 percentile range, while making sure to retain large elevation change events related to debris flows and rockfall (Table 2). Finally, the glacial elevation change over the entire period (2018-2013) is determined for a 20 m wide zone next to the proximal slope of the moraine, which is an indication of the glacial melt underneath the debris apron. When taking these corrections into account, the difference in elevation between two timesteps was assumed equal to the amount of sediment transport.

### 3.3 Analysis of surface properties

The orthomosaics were analyzed visually to examine patterns of erosion and deposition, and compared against elevation differences. Furthermore, the displacement, slope and roughness of the lateral moraines were derived from the DEMs. We employed the COSI-Corr software for cross-correlation feature tracking to calculate the displacement of debris on the lateral moraines (Leprince et al., 2007; Kraaijenbrink et al., 2016). As this software focuses more on block movement than on individual clast displacement (Leprince et al., 2007), correlating displacement with elevation change gives insight in slower slope processes such as creep or slow slumping, which often occur on a scale large enough to be detected by COSI-Corr. Fast events that travel

beyond our chosen maximum window size of 26.5 m, are registered by the algorithm as noise and hence do not bias our average velocities on the moraine. Slope maps are created directly from the DEM. Following Nield et al. (2013), the roughness length $z_0$ [m] was derived by

$$\ln(z_0) = 0.65 + 1.37 \ln(\sigma_z) \tag{1}$$

with $\sigma_z$ defined as the standard deviation of a $5 \times 5$ m window of a detrended DEM. Although the window size greatly influences the roughness, a window of 25 m$^2$ is suitable for this approach (Miles et al., 2017b). For each window a high roughness value indicates larger topographic variation, such as boulders, while a small value indicates a more homogeneous surface (Miles et al., 2017b).

### 3.4 Moraine delineation

The DEM and orthomosaic, as well as their derivatives, were used to delineate the lateral moraines and divide them into zones with comparable characteristics. The moraine base is often characterized by a break in slope (Figure 4, Figure 5B). Furthermore, a hillshade with a hummocky appearance is an indicator for subdebris ice (Lukas et al., 2012), and these areas are thus excluded from the moraine. Within the lateral moraine two main zones were distinguished: an intensively gullied upper part (Curry et al., 2006) and a lower part that consists mainly of reworked debris, usually in the form of coalescing debris cones, (Figure 2, Figure 4, Figure 5A), as also described by Ballantyne (2013). Although not distinguishable everywhere, a zone with fine material was detected directly below the gullied upper part, accompanied by a very low roughness (Figure 3B, Figure 5C). The smoothness is also visible on the orthomosaic. We interpret this part of the coalescing cones as the deposition zone of surface wash from the gullies upslope.

### 3.5 Runout model

To investigate the importance of the lateral moraines as a source for supraglacial debris, we used a simple model to calculate how far moraine material can travel onto the glacier. The model is based on the reach angle principle. The reach angle is the angle between the origin and maximum reach of a mass movement, and has a range between $3°$ and $45°$ (Evans and Hungr, 1993). For debris flows the reach angle is generally between $26°$ and $34°$ (De Haas et al., 2015). These variations are mainly caused by differences in processes, but the volume of the mass wasting is also important (Dai and Lee, 2002). Taking both process and volume into account, minimum reach angles for rockfall, shallow slides and debris flows are found to be $33°$ (with a volume of 100 - 1000 m$^3$), $23°$ (800 - 2000m$^3$), and $22°$ (800 - 2000m$^3$) respectively (Corominas et al., 2003). Reach angles decrease for values beyond these ranges, and increase for smaller volumes.

The runout length $R_L$ (m) from the lower moraine boundary was calculated as

$$R_L = \frac{\Delta H}{\tan(R_\alpha)} - M_w \tag{2}$$

where $\Delta H$ is the difference in elevation between start end location of deposition (m), $R_\alpha$ (°) is the reach angle (Figure 9A) and $M_w$ is the planar moraine width (m). To derive the maximum runout length, it was assumed that the start location of the mass transport is at the moraine crest, though in reality they may start from anywhere inside the gullied zone (Curry et al., 2006).

5 As the minimum reach angles were used and since we assume the moraine crest as a starting point, the calculated runout length $R_L$ indicates the furthest inward point on the glacier that debris can directly be transported to. Due to a decrease in mass movement velocity after the abrupt slope change on the glacier-moraine boundary, the amount of debris deposition is expected to be highest close to the moraine and will decrease rapidly with distance. To validate this estimated runout length, the actual runout length is determined by detecting depositional features such as debris flow lobes and rockfalls, and measuring their 10 distance to the moraine edge.

## 3.6 Clast analysis

Model results were validated by performing a clast analysis, which is used to distinguish between actively transported clasts and those that are mostly affected by weathering and reworking in rapid mass movement events (Lukas et al., 2013) and passively transported by the glacier. Moraine-derived debris is assumed to have already been actively transported by the ice during 15 moraine formation, when the subglacial sediment was being deformed. Therefore it has a higher roundness than passively transported rockwall-derived debris, which is expected to be more dominant in the centre of the tongue. If the lateral moraine indeed is an important source of debris, clast roundness is expected to decrease from the lateral moraine towards the glacier centre, as the influence of the lateral moraine diminishes and material transported from further up-glacier becomes dominant. The clast analysis was conducted by investigating 70 individual samples of debris. For each of the locations on average 46 ($\sigma$ 20 = 25) clasts were analyzed. The roundness is determined based on the commonly used chart from M. C. Powers (1953), which results in a percentage of clasts for each sample that are angular or very angular (RA index). For 13 samples the axis length of each clast was measured, allowing us to determine the so called $C_{40}$, which is the 40th percentile ratio of short to long axes (c/a) in a sample. Actively transported clasts are more likely to have low $C_{40}$ and RA values, in contrast to clasts that experienced passive transport (Lukas et al., 2013). It has to be taken into account that both the $C_{40}$ and RA indices decline down-glacier 25 (Benn and Ballantyne, 1994) and that differences in lithology result in different index values (Lukas et al., 2013). However, the latter will be of minor importance as the debris catchment is relatively small and homogeneous in lithology (Macfarlane et al., 1992).

## 4   Results and discussion

### 4.1   Observed surface lowering rates

30 The mean elevation change rate of the non-vegetated moraine between May 2013 and April 2018 equals -0.31 $\pm$ 0.26 m yr$^{-1}$ (Table 3). Most elevation change occurs on the lower moraine, consisting of extensively reworked coalescing debris cones, at

a rate of approximately -0.41 ± 0.21 m yr$^{-1}$. The glacier downwasting near the debris apron occurred at a rate of -0.60 ± 0.45 m yr$^{-1}$, which indicates a deposition rate of +0.19 m yr$^{-1}$ on the apron, assuming a similar downwasting rate below the apron. The upper gullied part on average has a surface lowering rate of -0.16 ± 0.26 m yr$^{-1}$. Debris remobilization on other moraines that formed during recent glaciations generally have lower elevation change rates than those observed on the gullies in this study, but also peaked at approximately -0.15 m yr$^{-1}$ (Curry et al., 2006), with Ballantyne and Benn (1994) noting their rates between -0.01 and -0.02 m yr$^{-1}$ to be minimum rates. The high gully erosion rates found are indicative for rapid surface change on steep lateral moraines above downwasted glacier tongues during deglaciation and debuttressing. The vertical accuracy of the dataset is determined by calculating DEM differences for off-glacier terrain for all datasets. The total offset is -0.05 ± 0.84 m over an area of 1.6 km$^2$. Although the offset varies from -0.04 ± 0.70 to 0.12 ± 0.66 m between the different timesteps, it is at any time smaller than the observed surface lowering rates (Table 4).

## 4.2   Mass transport mechanisms and processes

Our data enables us to differentiate between different transport mechanisms, which can be divided in three main categories: erosion due to running water (entrainment and debris flows), larger mass movements (slumps and rockfall) and slower downslope processes (for example slow slumping). Despite the stability of inter-gully arêtes on the upper moraine, the gullied topography indicates the importance of flow erosion processes. Debris flows and sediment loaded streams originate here mostly in the wet season, when there is frequent rainfall, often with high intensity (Table 3). Relatively high rates of surface lowering can be found just below the gullies, which might be related to the presence of easily erodible surface wash deposits (Figure 6A). The fact that these higher rates can be found below more intensely gullied upper sections support this. Another possibility is that these higher rates are caused by a steep scarp of fresh-looking till which commonly forms at the very base of the eroding till cliff. This scarp may be caused by the separation of the debris apron from the gullied upper section as a result of sub-apron glacial downwasting. However, no such sharp step is visible on the orthophotos, keeping the exact origin unclear.

Further down the debris apron the slope decreases and depositional features are observed, mostly related to debris flows, as distinct levees along a central flowpath and lobe-like features (Figure 6C). Frequently observed grain diameters of > 40 cm on the debris apron furthermore indicate the importance of debris flows over water flows (Iverson, 1997), as debris flows, in contrast to water alone, are able to transport boulders of such a dimension. Nonetheless, many smaller channels show the importance of water flows for further reworking the sediment. Together with debris flows that cover the debris apron, water flow reworks the debris and transports material onto the glacier. However, due to the large elevation loss on the near moraine glacier, it is likely that there is net deposition on the debris aprons, and only a limited amount of debris is transported further onto the glacier. This is in line with the matching gully erosion rates (-0.16 m yr$^{-1}$) and approximate debris apron deposition rates (+0.19 m yr$^{-1}$). The steep gullied upper slopes are also susceptible to larger mass movements, such as the occurrence of slumps and rockfalls. These processes are enhanced by oversteepening of the slope and cause locally high (>2 m) erosion rates (Figure 6A1). One large slump event was captured in our data (Figure 6A1-A2), deposited mostly on the debris apron, where its loose material is susceptible to continued reworking. The debris apron of the moraine is partly eroded by water flow, but also moves downslope as a whole as the glacier ice below melts (Figure 6B). There is a horizontal displacement towards the glacier

of 0.93 m yr$^{-1}$, with a 90$^{\text{th}}$ percentile of 2.01 m yr$^{-1}$. This is in the range of movement by solifluction and that of rock glaciers (Matsuoka, 2001; Frauenfelder et al., 2005). Despite the fact that the local climate would favor solifluction (Gruber et al., 2017), it is unlikely to be the main transporting mechanism as velocities speed up in summer. The lateral displacement is also higher than those typically related to creep of unfrozen debris (Kirkby, 1967), thus it is most likely caused by slow slumping of the debris apron. Although its contribution to surface lowering is unclear, this process does result in a steady debris supply to the glacier.

## 4.3   Temporal patterns in surface lowering

Moraine surface lowering occurs throughout the year, and a significant difference was observed between the wet (-0.39 $\pm$ 0.74 m yr$^{-1}$) and dry (-0.23 $\pm$ 0.68 m yr$^{-1}$) season. Elevation change rates ranged from -0.34 $\pm$ 0.57 to -0.52 $\pm$ 0.84 m yr$^{-1}$ in the wet season and -0.17 $\pm$ 0.44 to -0.36 $\pm$ 0.37 m yr$^{-1}$, in the dry season. This difference is caused by heavier precipitation during the summer months (540 - 700 mm) as during the winter months (117 - 145 mm) in addition to higher glacial melt rates of -0.76 $\pm$ 0.73 m yr$^{-1}$ and -0.57 $\pm$ 0.45 m yr$^{-1}$ respectively. On the upper gullied moraine, erosion rates were highly variable between the dry and wet season (-0.07 $\pm$ 0.43 to -0.28 $\pm$ 0.77 m yr$^{-1}$), but generally lower than rates on the debris apron of the moraine (-0.25 $\pm$ 1.0 to -0.60 $\pm$ 0.82 m yr$^{-1}$). The remarkably high elevation loss on the debris apron during the wet season is mostly linked due to high glacier melt rates in the summer season. In addition, more active mass transport processes were observed in the wet seasons, contributing to the higher surface lowering rates. This also indicates the importance of water-driven erosion and slumping (Table 3).

The upper gullied moraine experiences less change throughout the dry season, but its high seasonal differences indicates its sensitivity to precipitation changes, which affect debris flow probabilities. During the dry season from 2016 to 2017, when surface lowering of the gullied part was especially high, both the total precipitation as well as the intensity were considerably higher. An increase of debris flow related landforms after the wet season co-occurs with the larger negative elevation change that was observed on the gullied upper part. Larger slumps and rockfall from this section were not limited to a single season, as they are mostly triggered by a single rainfall event rather than by continuous wetting. Surface wash deposits below the gullies can also be seen throughout the wet and dry season, as is the case for slow slumping of the debris apron. Nonetheless, the rate of movement was much faster during the wet summer season, as a larger loading and a decrease of shear strength may cause the moraine to slump faster as a result of buried ice melt (Cai and Ugai, 2004). Although the exact nature of this process is yet unclear and needs further investigation, it is clear that this does contribute substantially to the downslope movement of moraine debris.

Beyond precipitation, freeze-thaw cycles could also play a role in driving erosion, with erosion increasing as moraine slopes warm up seasonally after the dry winter season and diurnally through the rest of the year.

## 4.4   Towards a conceptual lateral moraine mass transport model

The steep intensely gullied upper part of the moraine had lower elevation change rates (-0.16 m yr$^{-1}$) than the lower debris apron part (-0.41 m yr$^{-1}$) (Figure 6). However, with a glacial melt rate of -0.60 $\pm$ 0.45 m yr$^{-1}$ there is net deposition on the

debris apron. This is in line with other studies that indicate that post-deglaciation the steep upper slopes get rapidly deprived of their loose sediment (Ballantyne, 2013; Curry et al., 2006). This would suggest higher erosion rates on the upper moraine part in the past, and this hypothesis is supported by the large amount of deposited material below. Currently, erosion still occurs in the gullies albeit at a lower rate, and gully surface wash deposition just below the gullies fills the gaps between the larger

boulders and decreases the surface roughness locally (Figure 5C). These finer grained deposits are susceptible to reworking by water flow (van Rijn, 1984).

Debris flows originating in the upper gullies both deposit and entrain material on the coalescing debris cones that form the lower part of the moraine. In addition, mass transport on this section of the moraine is caused by a slower slumping process in the wet season, which moves down the moraine as a block. During the wet summer, when surface displacement rates are

on average $>1$ m yr$^{-1}$, higher velocities coincide with a more negative elevation change, possibly related to the melt of ice underneath (Figure 8). In the dry winter season, displacement rates are lower ($< 1$ m yr$^{-1}$) and do not show a correlation with elevation change, indicating a smaller importance of slow slumping processes. This is remarkable considering the still relatively high melt rates (-0.57 $\pm$ 0.45 m yr$^{-1}$). The slumping might be enhanced in close proximity of areas of high glacier ice mass loss, as the ice thins vertically but also recedes laterally (Figure 7). These slump deposits may be identified on the

moving glacier as positive surface elevation change due to the dispersal of the deposited slump sediments (Figure 7B). The gullied upper moraine did not show any horizontal displacement over both seasons, which indicates the absence of sub-moraine ice in these parts and the absence of a slumping process. As these displacements can be linked directly to a slumping process they can easily be used to indicate areas of active mass transport on lateral moraines.

It is difficult to quantify the importance of fast processes such as debris flows, rockfall of individual large boulders and land-

slides on the moraine. However, the intensely gullied upper part prevents large flows to occur. Though rockfalls and a single slump are observed, they are infrequent and occur at a small scale (Figure 6). As a result, large infrequent events are of minor importance for the lateral moraine-derived sediment budget on this glacier.

The importance of frost action for supplying detachable sediments can also not be directly derived from the DEM differencing. However, with surface temperatures well below freezing level for a long time in winter, as evidenced by field measurements,

the formation of ice crystals is likely. These may detach sediments that can easily be removed afterwards. No debris flows or landslides were originated on this part, hence it is most likely that mass transport on the lower part was triggered by water and debris flows that originated in the upper gullies. The coalescing debris cones that make up the lower part of the lateral moraine along this glacier are constantly reworked, as debris flows, water flow and slow slumping transport material to the glacier. However, net deposition takes place from the gullies above, in contradiction to what the elevation change rates suggest.

The continuous glacial melt also constantly increases the sediment accommodation space, resulting in less debris to reach the glacier surface.

## 4.5   Debris distribution onto the glacier

The main processes (debris flow, shallow slides, rockfall) described above were included in the model to calculate on-glacier debris deposition.

Using a specific reach angle for each process, the maximum runout length on the glacier is 39 m for rockfall, 111 m for debris flow and 122 m for shallow slides (Figure 9B). The runout length is not equal along the glacier, as a result of differences in moraine elevation and the hummocky glacier surface (Figure 9C). The observed runout length is manually derived from the imagery and has a maximum of 51 m, which suggests rockfall as the most important process, closest to this value. As rockfalls

were not observed to be the most important process on the moraine, this difference also indicates that debris flows and small slumps possibly occurred with smaller runout lengths than modelled. There are two possible explanations for this. First, many mass movements might have a smaller volume than the 800-2000 $m^3$ range used in the calculation, which reduces the runout length (Corominas et al., 2003; Rickenmann, 2005). Second, the rough surface on the glacier obstructs the runout path and decreases runout length substantially (Corominas et al., 2003; Miles et al., 2017b). Using a smaller volume (<800 $m^3$) and

an obstructed path, the reach angle of debris flows and shallow slides decreases to 30°, which results in a maximum runout length of 56 m, much closer to the observed length of 51 m (Figure 9B). Nonetheless, both the calculated and measured runout lengths indicate that debris from the moraine cannot directly reach the centre of the glaciated surface, which is approximately 200 m from the moraine. Rough calculations of required reach angles for moraine-derived debris to reach the centerline on other glaciers in the catchment, based on the mean moraine prominence (Miles et al., 2016), shows that this is likely true

there as well. For the largest glacier, Langtang, the reach angle becomes 6°, while for the second largest, Langshisha, 18°. Due to differences in valley shape and moraine size, very different rates of relative coverage by moraine-derived debris are however possible. Due to an increasing elevation range between the moraine crest and glacier tongue, the runout length has increased over time. The debris currently found at the glacier's centerline has either moved there by secondary processes such as glacier movement and on-glacier sliding, or originated from other sources, e.g. rockwalls (Benn and Owen, 2002) or basal

debris, (re-)emerging at the glacier surface towards the tongue (Boulton, 1978; Wirbel et al., 2018). This result also indicates that the proportion of moraine derived supraglacial debris increases as the glacier downwastes. The formation of a fully lateral moraine derived debris cover is more likely in case of upstream medial moraine formation by confluencing glaciers, from which secondary dispersal mechanisms may form a supraglacial debris cover (Kirkbride and Deline, 2013b). Furthermore, advancing glaciers will have little moraine derived input due to a lack of extensive lateral moraine slopes.

## 4.6   Clast analysis

The locations of the clast samples are shown in Figure 10. The $C_{40}$ index is relatively low (0.2 to 0.48), while the rounded and very rounded fraction of the investigated sample is less than 6% on average, indicative of angular clasts. Looking at the angular and very angular fraction, the RA index, provides a stronger indication of transport processes (Figure 10). The RA index on the moraine is on average just above 30% suggesting a dominance in more rounded samples. This is due to their previous transport

path along the glacier bed before they were deposited along the moraine. Values are decreasing downglacier, corresponding to findings in Benn and Ballantyne (1994), indicating that englacially transported and moraine-derived debris becomes dominant. Higher RA indices on the moraine are found only where the moraine is still connected to the rockwall (Figure 10). Clast samples in the centre of the glacier have a much higher RA index, >50%, as they have been dominantly sourced directly from rockwalls further up-glacier, and less frequently emerged from englacial pathways. Clast samples from between the moraine

and the modelled runout length have an RA index higher than 50%, closer to the RA index of the samples in the centre of the glacier. This may indicate that the modelled runout length is an upper maximum and debris from the moraines rarely reaches this far, which is in line with the observed runout length being shorter and the limited debris supply due to deposition on the debris apron.

## 4.7   Consequences for debris thickness, terminus retreat and glacier downwasting

Debris-covered tongues are schematized with a convex-concave up-glacier thickness pattern, causing lower tongues to be covered in thick debris that causes debris-covered tongues to stagnate (Anderson et al., 2018; Watson et al., 2016; Kirkbride, 2000; Kirkbride and Deline, 2013b). This can be explained with debris accumulating continuously at the snout at higher rates than it can be evacuated (Figure 11A2). Moraine-derived debris can not be directly deposited over the complete width of the tongue (Figure 11A1), although secondary processes may be capable to further distribute it across the glacier. However, to get a first order estimate of its potential relative contribution to the thick cover expected on the snout, surface lowering rates from the moraines were compared to the overall debris thickness of the glacier. These net rates of -0.16 m yr$^{-1}$ on the gullied upper part and +0.19 m yr$^{-1}$ on the debris apron suggest that no debris reaches the glacier surface, but only a relocation of moraine material takes place. However, we do see depositional features beyond the current debris apron on the glacier. In addition, one could argue that the glacial melt rates underneath the debris apron are slightly lower than those on the glacier due to the thick debris layer on top (Schomacker, 2008), which would decrease the net deposition rate on the apron and would at maximum result in no net deposition or erosion at the debris apron. This suggests that the maximum debris supply to the glacier is only related to the gully erosion rate of -0.16 m yr$^{-1}$, which occurs over an area of 0.095 km$^2$. If it is assumed that this amount of debris is distributed equally over the entire glacier surface within the UAV domain (0.33 km$^2$), this results in an annual increase in debris thickness of 0.05 m, although this is unlikely given the observed runout lengths. If the debris is only deposited within the area constrained by the maximum runout distance (0.19 km$^2$) this implies an annual increase in debris thickness of 0.08 m. Observed debris thickness for this glacier is in the range of 0.11 - 2.3 m ($\mu$ 0.84 m) (McCarthy et al., 2017) and 0.4 - 1.6 m closer to the moraine (Steiner et al., 2018) measured around the on-glacier weather station . Furthermore, on the lower part of the Baltoro Glacier in the Karakoram similar debris accumulation rates have been found, ranging between 0.05 and 0.30 m yr$^{-1}$, suggesting that debris-covered tongues get buried rapidly (Gibson et al., 2017b).

The clear concave arcuate terminus appearance of retreating DCGs as opposed to the generally convex terminus of clean glaciers however does suggest extensive moraine sediment supply, along with internal ablation due to drainage conduits. As the tongue retreats, the debuttressing of the glacier causes the moraine to slump and fill the space available with moraine material (Figure 11A3 and B2), which might cause the higher debris supply here. At Lirung Glacier, the terminus retreated at a rate of 30 m yr$^{-1}$ at the centreline and much slower at the margins as here the moraine-derived debris quickly covered the ice, resulting in the moraine crest to slump and become shallower (Figure 11B2), and melt rates to decrease.

It is also remarkable that the near-moraine glacier downwasting rate (-0.6 m yr$^{-1}$) is much lower than previously found downwasting rates over the entire glacier (< -1.3 m yr$^{-1}$ & -2.18 m yr$^{-1}$, Nuimura et al. (2017); Immerzeel et al. (2014a)). This indicates a possibly thicker debris cover near the moraines suppressing downwasting rates, which is also in line with the

lack of ponds and ice cliffs close to the moraine (Immerzeel et al., 2014a), and supports the debris supply rates presented in this paper. Considering these processes, our results indicate that lateral moraine mass transport can play an important role in debris supply to the margins of a downwasted glacier tongue with steep lateral moraines, where it offsets the downwasting of the ice with deposition of debris. As the glacier downwastes, more moraine surface is susceptible to erosion, while more space becomes available for debris apron formation, continuously inhibiting debris to reach the glacier surface. The lower the downwasting rate, the higher the possible surplus of debris that can possibly move beyond the apron and closer to the center of the tongue, resulting in a thickening debris cover and making it a negative feedback effect. This is important to take into account in energy balance models (Carenzo et al., 2016; Rowan et al., 2015), which often use a uniform debris cover derived solely from rockwalls and do not take deposition and re-mobilization of lateral moraine debris into account. Lateral moraine debris supply is also found to be important for the form of terminus retreat.

## 5 Conclusion

In this study a time series of five years of UAV data is used to investigate the importance of lateral moraine mass transport to a debris-covered tongue and the following key conclusions are drawn:

- The surface lowering on the lateral moraines is high at an average rate of 0.31 m yr$^{-1}$, also attributed partially to the melt of sub-moraine ice. This translates to a maximum increase of debris thickness of 0.08 m yr$^{-1}$ in a narrow runout zone of approximately 50 m next to the moraine.

- As the downwasting rate decreases, this rate of thickening will likely increase, resulting in a negative feedback with increasing debris thickness and reduced melt rates. If supplied far enough upstream, this additional debris may then be redistributed on the glacier surface.

- There is a strong seasonality in lateral moraine mass transport and the rates are higher in the wet summer season, which indicates the combination of water driven processes and glacier melt as key mechanisms.

- The steep upper part of the moraine is intensely gullied and transports debris quickly to the depositional debris apron. Here the more gentle slopes are reworked by water flow, debris flow entrainment and slow slumping.

- Rockfall and landslides occur occasionally and influence the vertical elevation change pattern, they are however of minor importance in the overall balance.

- Runout distance modelling shows that it is unlikely that these processes are responsible for the distribution of the eroded material on the glacier since the maximum distance is small (~56 m). This is supported by the clast analysis, which shows angularity to increase rapidly from the moraine (30% being angular or very angular) towards the glacier centre (>55% being angular or very angular), in addition to reduced melt rates near the glacier margin.

– Considering these results and revisiting our initial hypothesis, debris supply from lateral moraines alone can not explain the thick and continuous debris cover of tongues on debris-covered glaciers. While a considerable amount of debris from the moraines reaches the surface, it can only explain thickening on the margins of the tongue.

– Further research is needed that incorporates glacier dynamics and lateral drag with the moraines and its implications for possible debris transport on the glacier surface. In addition, methods to quantify rockwall erosion rates as well as subglacial erosion on debris-covered glaciers need to be developed to understand the full sediment balance of a debris-covered glacier tongue. Better estimations of sub-debris apron glacier melt would also greatly improve the debris supply estimations. Using different future scenarios of glacier recession, the long term development of lateral moraines and their changing contribution to an increasing debris cover could also be investigated in a dedicated modelling study.

*Author contributions.* JS, PK, EM and WI developed the research goal. TW, JS and PK performed the primary data analysis and model development. TW, JS and WI wrote the manuscript. PK and EM helped with interpretation of results.

*Competing interests.* The authors declare no competing interests.

*Acknowledgements.* This project was supported by funding from the European Research Council (ERC) under the European Union's Horizon 2020 research and innovation program (grant agreement no. 676819) and by the research programme VIDI with project number 016.161.308 financed by the Netherlands Organisation for Scientific Research (NWO).
The input from Tjalling de Haas on mass transport processes is also greatly appreciated.

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

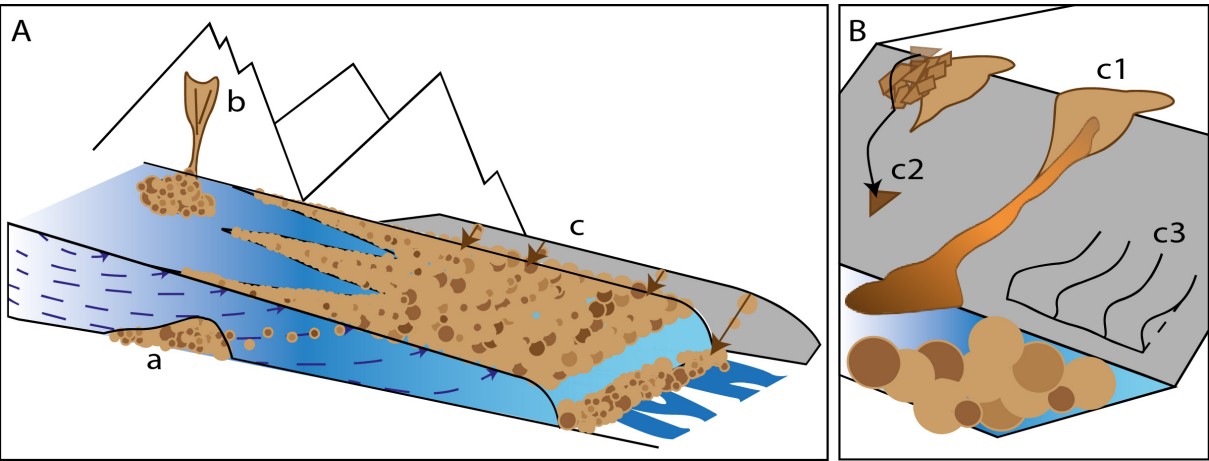

**Figure 1.** Schematization of debris supply processes towards the glacier surface (A), showing basal erosion (a), rockwall erosion (b) and sediment supply from lateral moraines (c). Mass transport processes on a lateral moraine bordering the glacier (B), showing debris flows (c1), rockfall (c2) and slow slumping (partly caused by the melt of sub-moraine ice (c3).

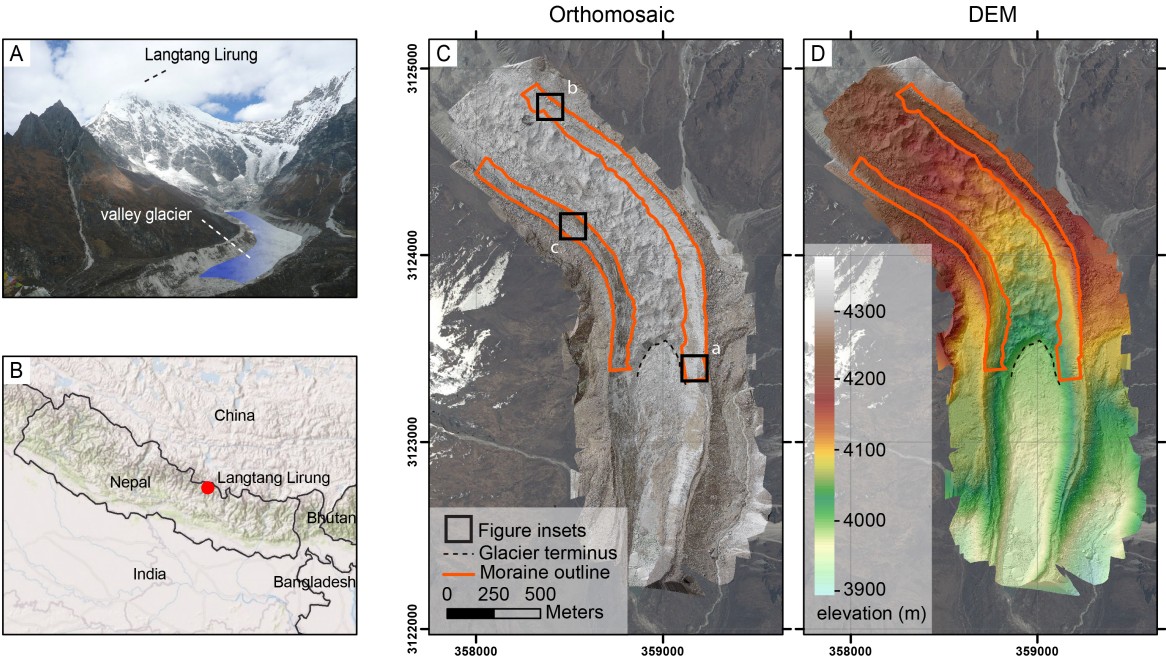

**Figure 2.** Study area including the setting (A), location of the study area (B) and two more detailed overviews including the orthomosaic and elevation data of the study area (C,D; from UAV flights in October 2015). The glacier terminus and studied section of the moraine are outlined on these maps. The white squares indicate the location of Figure 5 (a), Figure 6A (b) and Figure 6B (c).

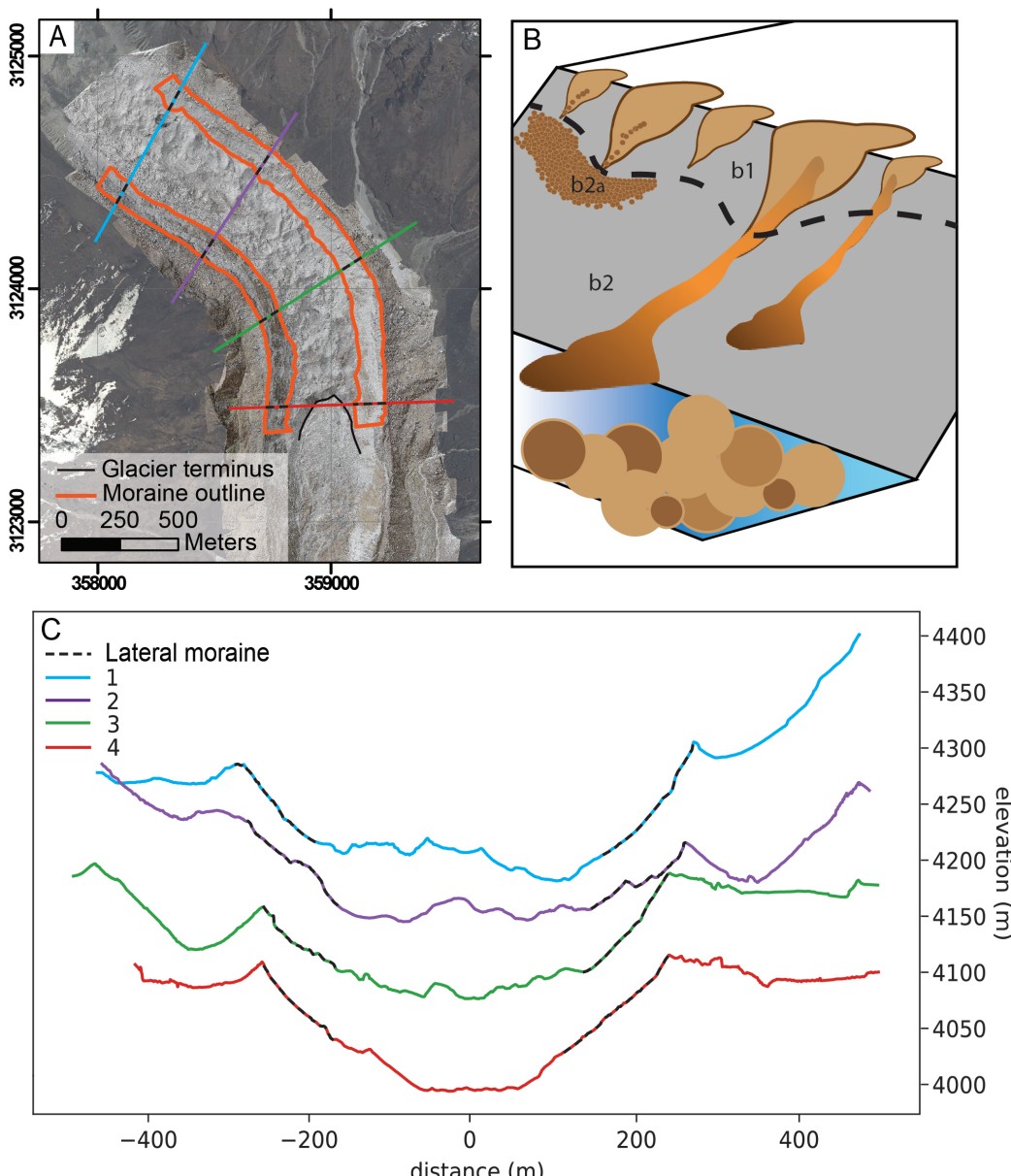

**Figure 3.** Cross sections over the glacier and moraine, with their locations (A) and corresponding elevation profile (C). They clearly indicate that the distal slope of the moraine has an opposite aspect to the proximal slope, due to which there is no debris input from higher slopes and rockwalls in the valley. In B the different sections of the moraine are indicated, with the upper gullied part (b1), the debris apron or coalescing debris cones (b2) and the smooth section of the debris apron interpreted as gully washout deposition (b2a).

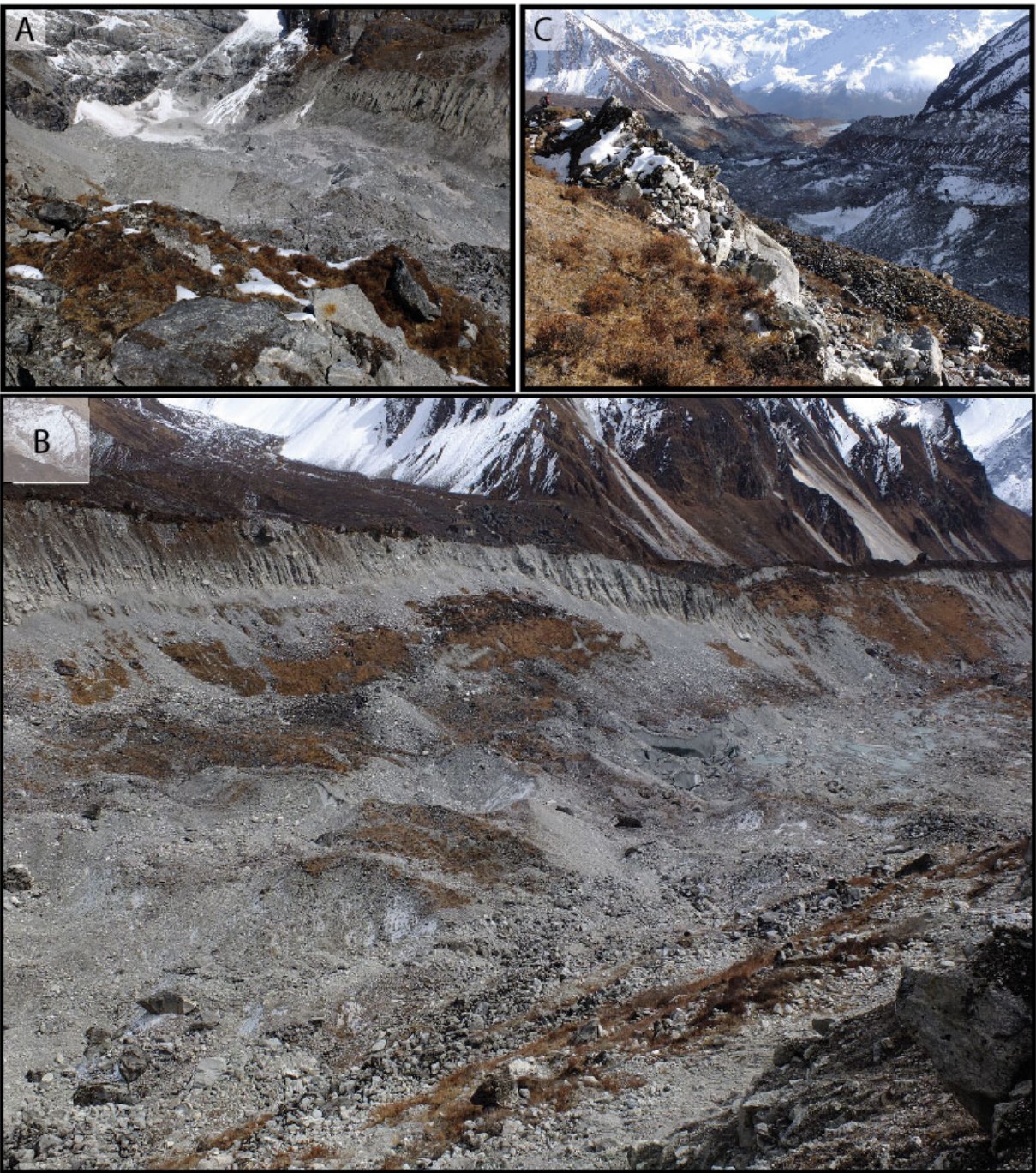

**Figure 4.** Field images of Lirung Glacier from the upper part (A), where the rockwalls are still connected to the glacier, along the middle disconnected part (B) to the lower glacier tongue (C), where the decoupling is clearly visible. Also note the mass movements from the upper slopes (B) that do not reach the glacier.

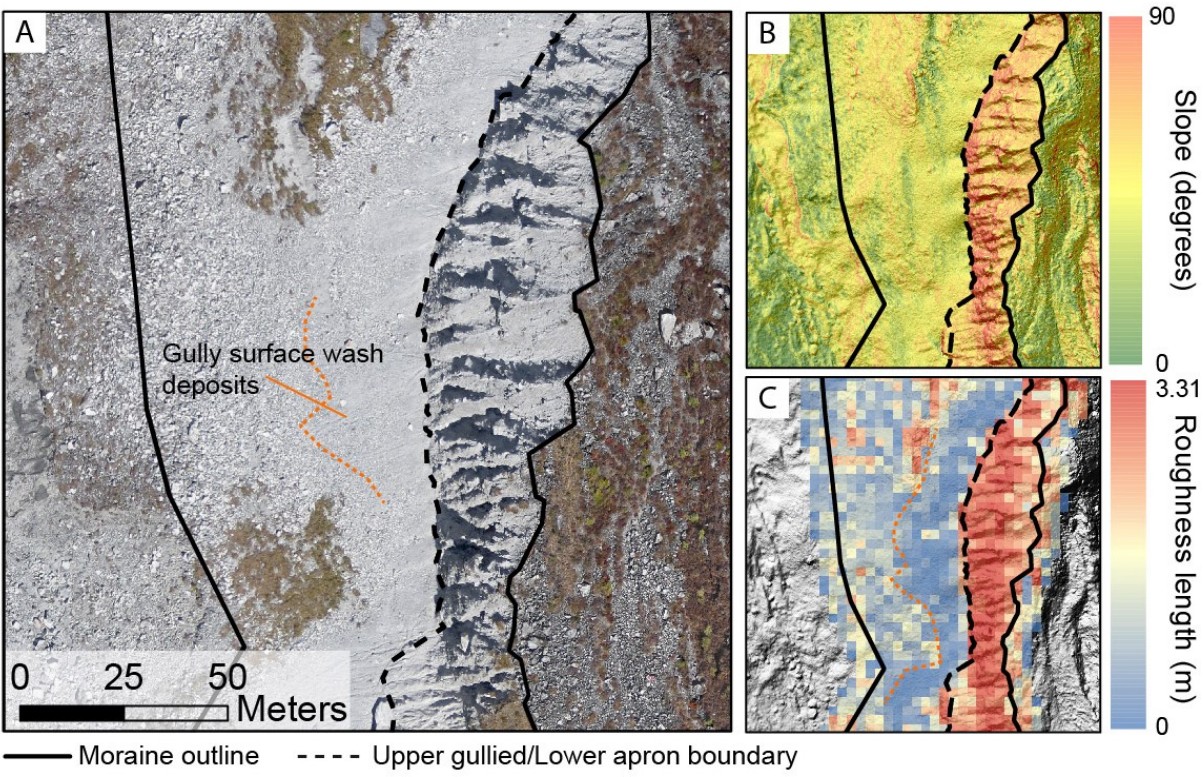

**Figure 5.** Moraine delineation method. The crest is often well defined by the orthomosaic (A) and slope (B). An intermediate zone with gully surface wash deposition is clearly visible from the roughness data (C). The moraine consists of an intensely gullied upper part (right) and a lower debris apron consisting of coalescing debris cones (left). See Figure 2 for location.

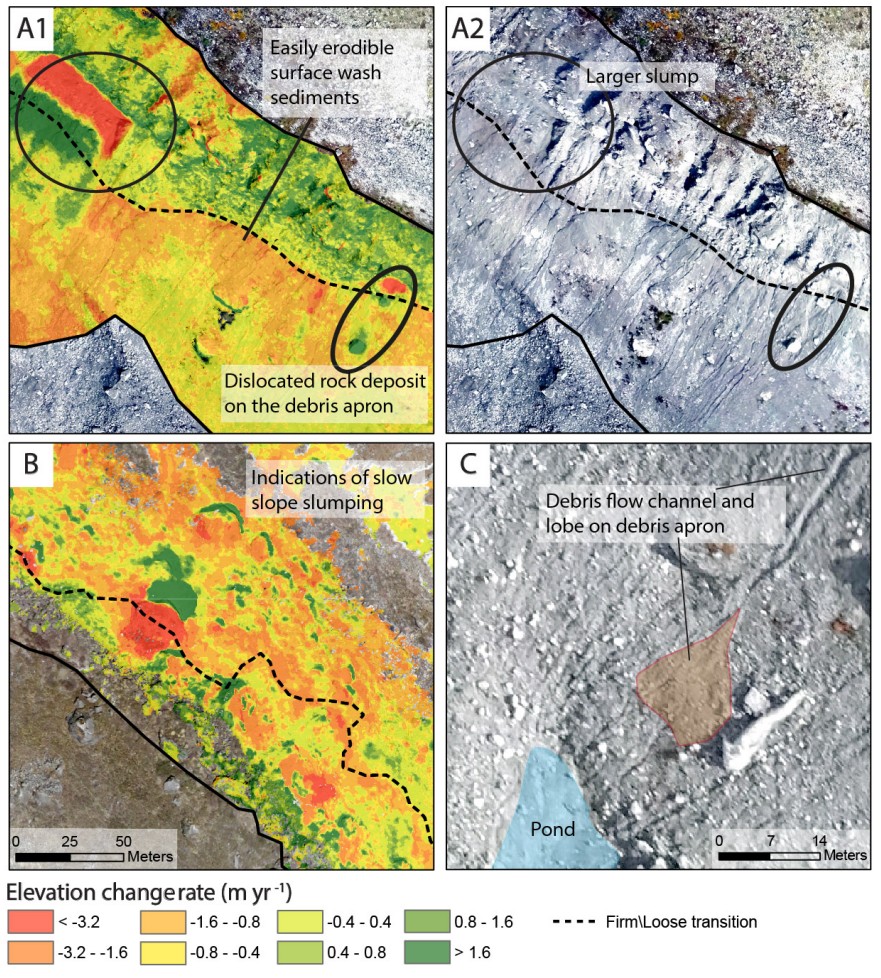

**Figure 6.** Different types or mass transport processes. Fast mass movements as slumps and rockfall occur on the moraine (A1, A2) as well as water driven movement as water flow and debris flows (C). Furthermore the slopes are slumping down slowly (B), which is clearly visible by the alternation of positive (rocks moving in) and negative (rocks moving out) elevation change values. See Figure 2 for location.

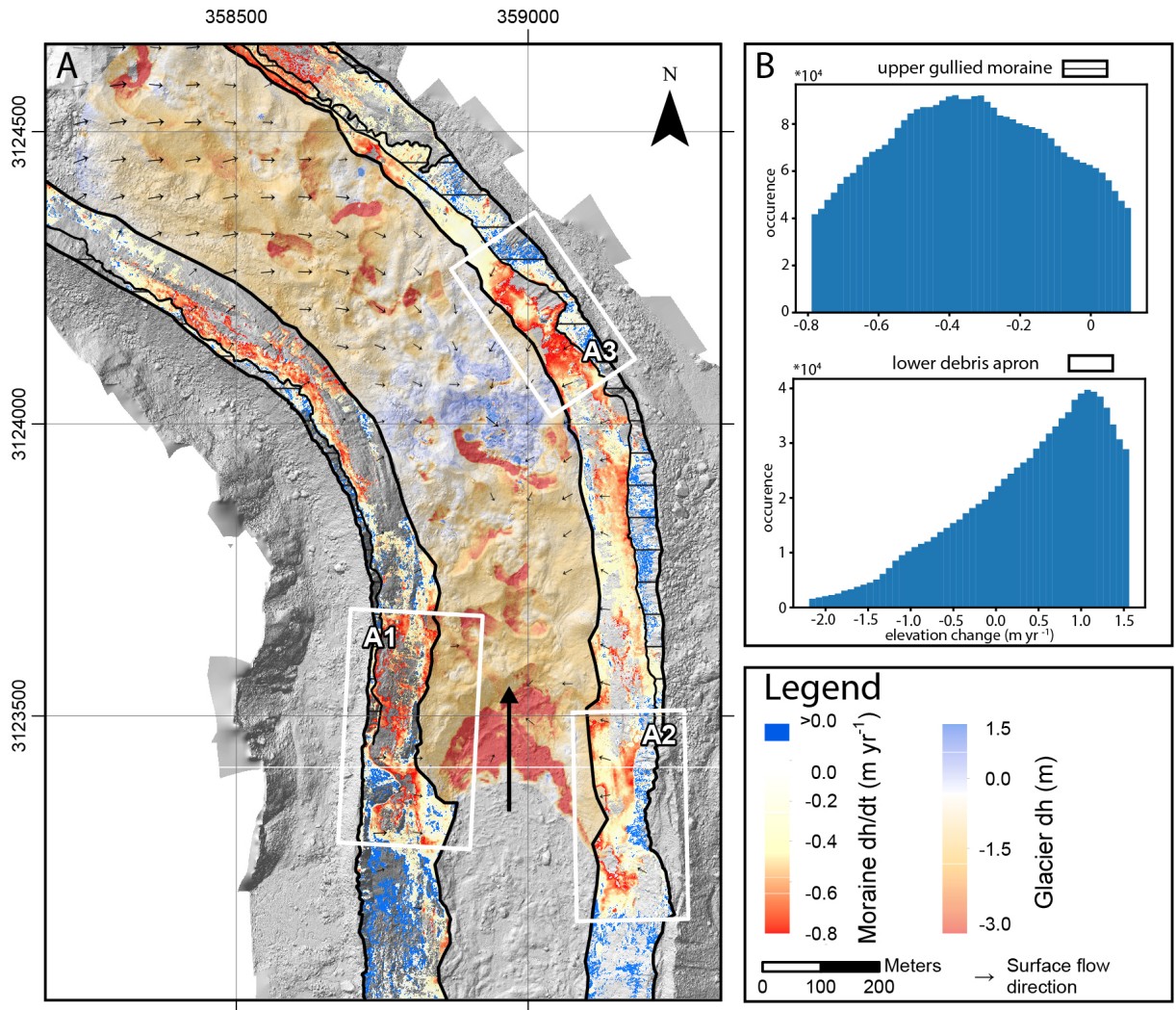

**Figure 7.** Downwasting of the glacier surface between May and October 2013 (Immerzeel et al., 2014a) and surface flow direction (Kraaijen-brink et al., 2016) compared to elevation change on the moraines (A). Insets A1, A2 and A3 show hotspots of surface lowering. Small black arrows show surface motion, either by active debris transport or passive transport imposed by the glacier velocity. The arrow size indicates exaggerated intensity of velocity, as velocities on the moraines (mean: 0.89 m yr$^{-1}$) are far smaller than on the upper glacier (mean: $\sim$ 6 m yr$^{-1}$). The long black arrow shows retreat of the terminus between 2013 and 2018 (~150 m). Blank areas are either outside the study area, vegetated or outside the 10-90 percentile range. (B) shows the histograms of the elevation change on the upper gulied section and lower debris apron over the entire studied period (2018-2013).

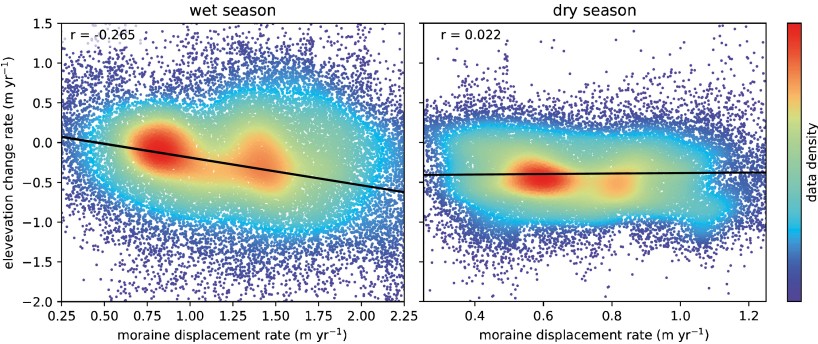

**Figure 8.** Relation between elevation change and moraine velocity in the wet and dry season. In the wet season, high velocities occur with higher surface lowering rates. Debris apron displacement is likely caused by a slumping movement due to sub-apron ice melt. No such relation exists in the dry season.

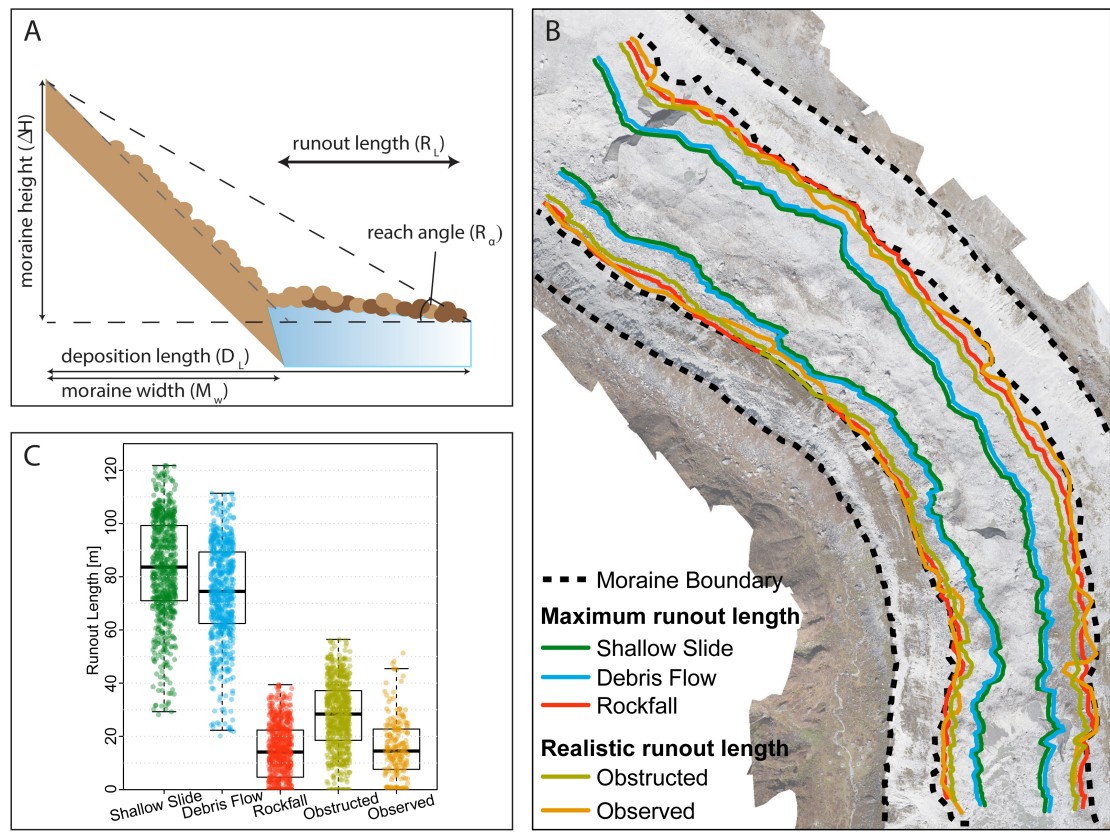

**Figure 9.** Conceptual diagram of model approach (A), the modelled runout lengths (B) and the modelled variability in runout lengths (C).

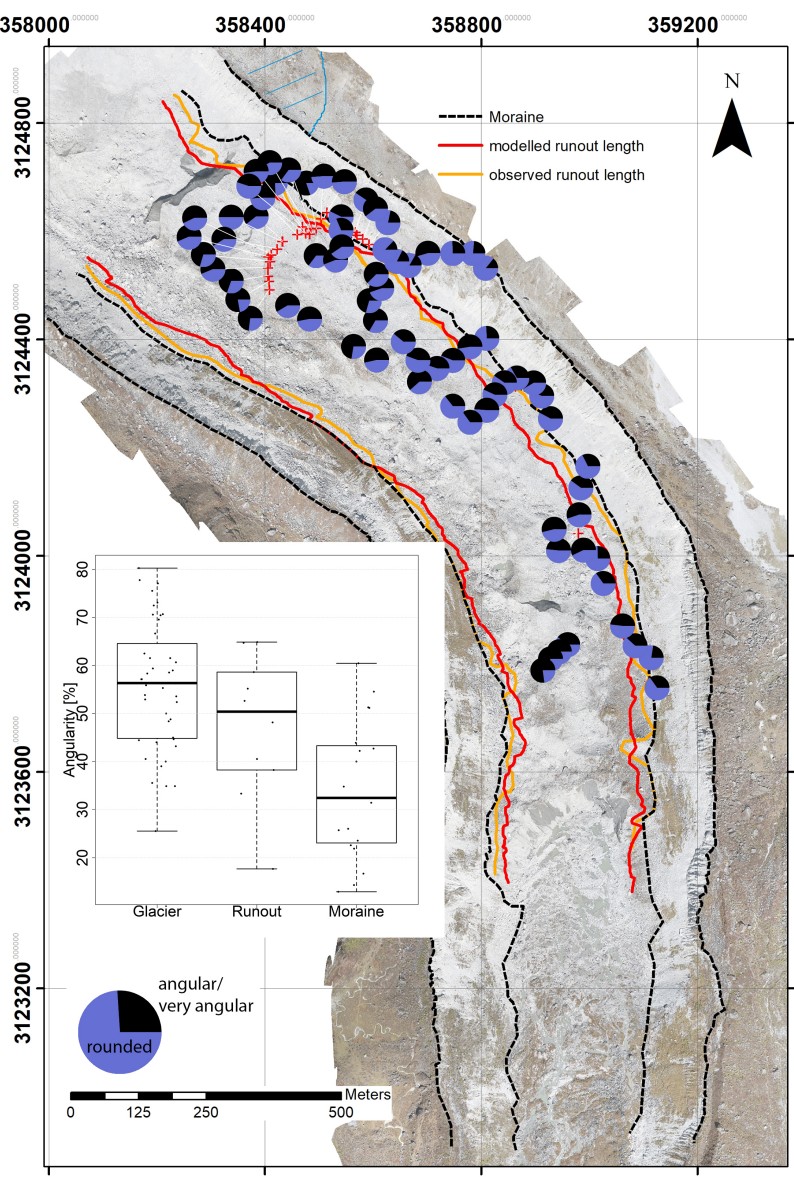

**Figure 10.** Angularity samples taken on the glacier surface, observed and modelled runout lengths are also shown. The inset shows angularity values for samples beyond the modelled runout length ('Glacier'), between the runout length and the base of the moraine ('Runout') and on the moraine ('Moraine'). In the top right corner, marked in blue, the glacier is still connected to rockwalls by way of avalanching.

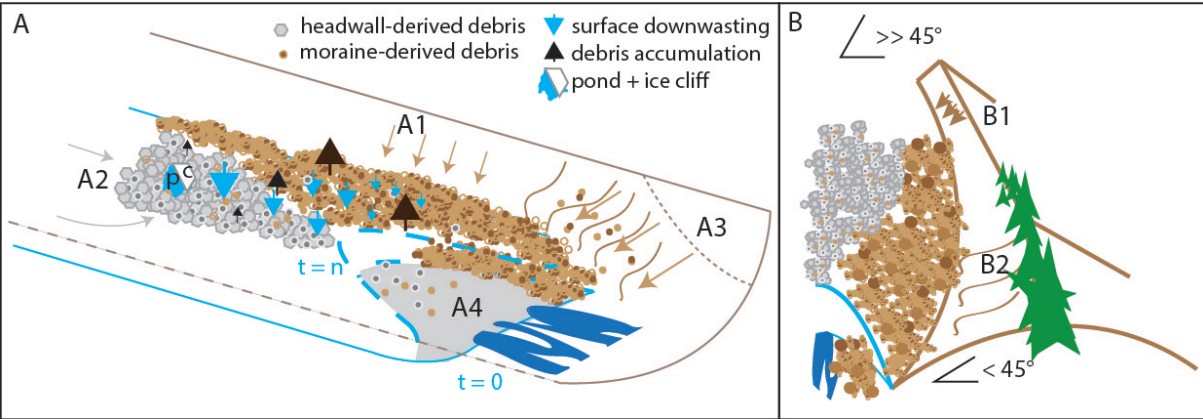

**Figure 11.** Dominant processes at lateral moraines and consequences for the glacier terminus of a debris covered glacier. Mass transport from lateral moraines (A1) as well as debris transport from head walls and englacial transport (A2) are shown. Slumping as a consequence of debuttressing at the terminus (A3) brings more material into the glacier forefield (A4), as the tongue retreats in time (t). As a consequence the moraine crest slumps (B2) and the moraine becomes shallower compared to upper parts where transport is mainly to rockfall and debris flows (B1).

**Table 1.** Date of acquisition and area of the different datasets.

| Date | mapped area [km$^2$] |
|---|---|
| 2013 / 05 / 18 | 2.04 |
| 2013 / 10 / 22 | 3.25 |
| 2015 / 10 / 18 | 2.61 |
| 2016 / 04 / 30 | 1.97 |
| 2016 / 10 / 06 | 2.70 |
| 2017 / 04 / 20 | 1.41 |
| 2017 / 10 / 19 | 2.32 |
| 2018 / 04 / 28 | 2.12 |

**Table 2.** Extreme values below and above which the data is removed from the original dataset, as well as the mean of the whole dataset ($\mu$).

| Dataset | $\mu$ [m] | $10^{th}$ percentile [m] | $90^{th}$ percentile [m] |
|---|---|---|---|
| 2013 / 05 - 2013 / 10 (wet) | -0.39 | -2.07 | 1.18 |
| 2015 / 10 - 2016 / 04 (dry) | -0.17 | -1.06 | 0.72 |
| 2016 / 04 - 2016 / 10 (wet) | -0.34 | -1.47 | 0.87 |
| 2016 / 10 - 2017 / 04 (dry) | -0.36 | -1.11 | 0.37 |
| 2017 / 04 - 2017 / 10 (wet) | -0.52 | -2.16 | 1.11 |
| 2017 / 10 - 2018 / 04 (dry) | -0.22 | -0.88 | 0.13 |
| 2013 / 05 - 2018 / 04 (total) | -0.31 | -0.31 | 0.79 |

**Table 3.** Seasonal elevation change values, furthermore divided in upper and lower moraine. The zonal mean ($\mu$) is reported, as well as the 1-sigma standard deviation ($\sigma$). Precipitation is measured at Kyanjing station in 2013 and Langshisha station in all others seasons. Elevation change values are in m yr$^{-1}$, precipitation values in mm and mm hr$^{-1}$. The significance column indicates the periods from which that specific dataset statistically differs ($p < 0.05$).

| Period | Dataset | Entire moraine | | | Gullied upper part | | | Debris apron | | | Precipitation | |
|---|---|---|---|---|---|---|---|---|---|---|---|---|
| | | $\mu$ | $\sigma$ | sig | $\mu$ | $\sigma$ | sig | $\mu$ | $\sigma$ | sig | cumulative | mean intensity |
| 1 | 2013 / 05 - 2013 / 10 (wet) | -0.39 | 0.79 | all | -0.28 | 0.77 | all | -0.41 | 0.83 | 2, 4-6 | 697 | 0.90 |
| 2 | 2015 / 10 - 2016 / 04 (dry) | -0.17 | 0.44 | all | -0.07 | 0.43 | all | -0.21 | 0.44 | all | 145 | 0.62 |
| 3 | 2016 / 04 - 2016 / 10 (wet) | -0.34 | 0.57 | all | -0.14 | 0.55 | all | -0.41 | 0.59 | 2, 5-6 | 584 | 0.58 |
| 4 | 2016 / 10 - 2017 / 04 (dry) | -0.36 | 0.37 | all | -0.24 | 0.37 | all | -0.41 | 0.36 | 1-2, 5-6 | 172 | 0.84 |
| 5 | 2017 / 04 - 2017 / 10 (wet) | -0.52 | 0.84 | all | -0.19 | 0.82 | all | -0.60 | 0.82 | all | 541 | 0.58 |
| 6 | 2017 / 10 - 2018 / 04 (dry) | -0.22 | 0.98 | all | -0.12 | 0.98 | all | -0.25 | 1.00 | all | 117 | 0.50 |
| 7 | 2013 / 05 - 2018 / 04 (total) | -0.31 | 0.26 | all | -0.16 | 0.26 | all | -0.41 | 0.21 | 2, 5-6 | 2257 | 0.67 |

**Table 4.** Off-glacier elevation differences on stable terrain. These differences can be seen as the vertical accuracy of that specific dataset. Values are in m.

| Dataset | $\mu$ | $\sigma$ |
|---|---|---|
| 2013 / 05 - 2013 / 10 (wet) | -0.04 | 0.44 |
| 2015 / 10 - 2016 / 04 (dry) | -0.03 | 0.74 |
| 2016 / 04 - 2016 / 10 (wet) | 0.12 | 0.66 |
| 2016 / 10 - 2017 / 04 (dry) | -0.02 | 0.69 |
| 2017 / 04 - 2017 / 10 (wet) | -0.04 | 0.70 |
| 2017 / 10 - 2018 / 04 (dry) | 0.08 | 0.93 |
| 2013 / 05 - 2018 / 04 (total) | -0.05 | 0.84 |