# Peer review of "Sediment supply from lateral moraines to a debris covered glacier in the Himalaya"

_Earth Surface Dynamics, 2018_

## Short Comment (SC1) · 26 Oct 2018

Referee Comments Earth Surface Dynamics Discussions Manuscript #: ESURF-2018-63

Title: Estimating lateral moraine sediment supply to a debris-covered glacier in the Himalaya

Authors: Teun van Woerkom, Jakob F. Steiner, Philip D.A. Kraaijenbrink, Evan S. Miles, and Walter W. Immerzeel

General Comments: The authors seek to quantify the amount of erosion occurring

on exposed lateral moraines from a Himalayan glacier and the characterize the redistribution of said material onto the modern glacier surface. This particular feature is an important aspect of glacier mass balance, especially in the Himalaya where debris covered glaciers are common. The authors utilize high resolution digital elevation model (DEMs) collected from unmanned aerial vehicles (UAVs) between May 2013 and March 2018 to quantify changes in moraine elevation which are inferred as equal to the amount of sediment transport from the moraines to the adjacent glacier surface. They conclude that lateral moraines are an important source of debris to the tongues of retreating debris covered glaciers. I applaud the authors on a simple, yet elegant, use of high resolution DEMs and clast observations to characterize the evolution of lateral moraines at an individual glacier and in an attempt to determine the contribution of the moraines to glacier debris cover. Overall the text of the manuscript is written clearly and organized in a logical manner, the methods utilized are, in general, well described. However, I do have some outstanding concerns regarding assumptions made by the Authors, DEM differencing, and the significance of the presented data, detailed below.

Specific Comments: • The Authors allude (in the abstract in on pg. 3, ln 2) to the fact that debris input from lateral moraines is more important for retreating glaciers or for glaciers with stagnant tongues. What is not explicitly stated is that contribution from lateral moraines only occurs when the surface of the glacier is below the crest of the lateral moraines; though this is partly common sense, I think that explicitly stating this in the introduction is worthwhile. It may also be pertinent to expand upon the temporal variability with regards to debris input; an advancing glaciers will have a higher proportion of debris sourced from the headwall, while a retreating glacier can accept debris input from lateral moraines. The authors speak to a this later in the discussion, but setting the stage in the introduction may better lead into the later discussion.

• Section 2 states that the moraines are 'disconnected' from upper slopes. I take this to mean that the outboard face of the lateral moraines have an opposite aspect as the valley wall (i.e., debris falling from the valley walls will likely not travel over the

moraine onto the glacier). Perhaps obvious to persons on the ground observing the glacier system, but explicitly stating this is important for justifying no debris input from valley walls. Also, comparing Fig. 1 to current google earth imagery, it is not clear if the upper part of the mapped moraines are actually disconnected from the valley wall.

• After referring to the potential for ice cored moraines potentially contributing mass as they degrade over time in the introduction, the Authors assume that no melting ice core exists in the moraine (such that all elevation changes are due to sediment transport off the moraines; pg. 3, ln 10). In a seemingly contradictory statement, the authors discuss a '...hillshade with a hummocky appearance is an indicator for sub-debris ice' (pg. 5, ln 4) when discussing the lateral moraines. Ice cored moraines are generally quite commonly associated with debris covered glaciers (Clark et al., 1994), and this catchment is no exception. The authors provide no observations or other evidence to support their assumption that the moraines are not ice cored. Ice cores within moraines are known to persist for thousands of years and can help maintain steeper moraine slopes (Crump et al., 2017), which are present in the Lirung glacier. Visual observations from current Google Earth imagery shows that appears to be a consistent debris cover across most of the glacier and pro-glacial area, which suggests that the glacier very likely could have formed ice cored moraines in the past. Without concrete evidence that the existing lateral moraines are not significantly ice cored, I am hesitate to agree with the assumption that all elevation changes on the moraines are solely due to sediment transport.

• In section 4.1 a vertical error of 0.02 $\pm$ 0.33 m y-1 is quoted, citing (Immerzeel et al., 2014). Upon further reading of (Immerzeel et al., 2014), it is difficult to tell, but it appears that the off-glacier area used to compute this error is actually the lateral moraines themselves, which are likely to change in elevation between May and Oct 2013); perhaps I am interpreting this incorrectly, but moraines don't seem like an ideal location to compare the accuracy of DEMs due to their changing elevation. Maybe the vegetated areas are more stable, so are appropriate to use to constrain DEM accuracy

at different timesteps? Also the quoted accuracy of the DEMs is only calculated for DEMs produced in May and Oct of 2013; what about the accuracy of DEMs produced up through 2018? Further reading of the more recent Kraaijenbrink et al. (2016) indicates that "The bulk of the vertical errors at the tie points are within 50cm, and 75% are even within ~25cm." I am by no means an expert in DEM creation, let alone from high resolution UAV imagery, but if this is the actual uncertainty in elevations between DEMs, then the reported elevations changes in this manuscript (e.g., Table 3) are well within this uncertainty. Therefore, how can you be certain that your extracted measurements are above the vertical uncertainty and noise between your DEMs from different times? Perhaps I am not reading Kraaijenbrink et al. (2016) correctly, but at minimum, it would be appropriate to add more detail regarding the vertical errors associated with your DEMs into this study (in addition to citing Immerzeel et al. (2014) and Kraaijenbrink et al. (2016)), since changes in elevation are the key component of this study. The Authors mask out vegetation areas from the DEMs to ensure that the moraine elevations changes represent real change. I am confused by this. Mass wasting events and sediment transport within vegetated areas aren't real change? Perhaps I am not interpreting this correctly. The Authors also state that they correct the DEMs for off-glacier and off-moraines elevation changes with the assumption that those landscapes should be stable. Again it is not entirely clear why the DEMs must be corrected for these large changes; aren't large sediment transport events possible on 'stable' landscapes (though less common)?

• Table 3 presents the elevation changes derived from differencing various DEMs over the study period. I am assuming that the mean the 1sigma uncertainty is presented, but this is not explicitly stated in the table (see technical corrections). When plotted, the mean and uncertainties all overlap well within each other's (see attached Fig. 1). Are these measurements statistically different from each other? Enough to back up the arguments regarding the spatial and temporal patterns presented in the manuscript? It would be nice to see the raw data plotted in a histogram to see the distribution of elevation changes within each different moraine region/time period (e.g.,

Immerzeel et al. (2014) Fig. 6). This would allow the reader to get a better sense for how the calculated elevation differences vary within a region and a nice accompaniment to Fig. 6.

Technical Corrections: • Please see attached PDF for technical corrections.

Overall Comments: I applaud the authors for tackling a complex problem that is nonetheless very pertinent for understanding the dynamics of debris covered glaciers. Using higher resolution DEMs to investigate many aspects of moraine evolution through space and time is an important task. However, I have a few concerns that I believe should be addressed before this manuscript moves forward towards publication. First, the assumption of no ice-cored moraines is substantial, and one that should be backed-up with field observations or other evidence. Debris covered glaciers are notorious for producing ice cored moraines, and elevation changes triggered by melt-out could be an important component. Also, the large variability of reported elevation change measurements and the overlap between measurements makes delineating significant differences between measurements difficult (see Fig. 1 above). A more detailed treatment of variability in reported elevation changes would greatly benefit the manuscript.

  References Clark, D.H., Clark, M.M., and Gillespie, A.R., 1994, Debris-Covered Glaciers in the Sierra Nevada, California, and Their Implications for Snowline Reconstructions: Quaternary Research, v. 41, no. 2, p.139–153, doi:DOI: 10.1006/qres.1994.1016. Crump, S.E., Anderson, L.S., Miller, G.H., and Anderson, R.S., 2017, Interpreting exposure ages from ice-cored moraines: a Neoglacial case study on Baffin Island, Arctic Canada: Journal of Quaternary Science, , doi:10.1002/jqs.2979. Immerzeel, W.W., Kraaijenbrink, P.D.A., Shea, J.M., Shrestha, A.B., Pellicciotti, F., Bierkens, M.F.P., and De Jong, S.M., 2014, High-resolution monitoring of Himalayan glacier dynamics using unmanned aerial vehicles: Remote Sensing of Environment, v. 150p.93–103, doi:10.1016/j.rse.2014.04.025. Kraaijen-brink, P., Meijer, S.W., Shea, J.M., Pellicciotti, F., De Jong, S.M., and Immerzeel,

ESurfD
W.W., 2016, Seasonal surface velocities of a Himalayan glacier derived by automated correlation of unmanned aerial vehicle imagery: Annals of Glaciology, v. 57, no. 71, p.103–113, doi:10.3189/2016AoG71A072.

Please also note the supplement to this comment:
https://www.earth-surf-dynam-discuss.net/esurf-2018-63/esurf-2018-63-SC1-supplement.pdf

[Figure]

Figure 1: Data from Manuscript Table 3 plotted (dots = μ and error bars = σ).

**Fig. 1.** Plotted Data from Manuscript Table 3

**Supplement:**

[revised manuscript text omitted]

---

## Short Comment (SC2) · 26 Oct 2018

Apologies to both the Editors and the Authors! This was supposed to be submitted as a RC, not an SC (confusion on my end).

Happy to discuss any and all of my comments, please do not hesitate to reach out! Regards, Simon

---

## Referee Comment (RC1) · M. Truffer (Referee) · 2 Nov 2018

This paper shows a very unique data set with several repeat DEMs that were acquired over highly erodible lateral moraines. These data allow a quantification of the debris flux from such moraines to the glacier surface, where they influence the thickness of the supraglacial debris layer and thus the glacier's mass balance. The paper is highly relevant and I recommend publication. There are several opportunities to make it clearer, however, and I recommend a number of revisions.

Overall, there are two issues that should be addressed clearer in the paper:

[Figure]

1) In the paper erosion from several sources (headwall, lateral, basal) is treated similarly and independently. But this is not really the case. All sediment in lateral moraine has originated from headwall erosion, transported by the glacier and is now remobilized. It's not primary erosion. So it doesn't really make sense to compare rates, these are two different things. It would improve the paper greatly if this distinction was clearly made in the Introduction as well as the Discussion. This would also outline some of the issues that need to be addressed in coupled debris/ice flow models, namely sediment deposition and re-activation.

2) The derived rates are based on the assumption that none of the elevation changes on the moraine are due to melting ice. This seems like a big assumption and it is only mentioned in one sentence and no potential errors due to this assumption are discussed.

Detailled comments:

p.1, l.5: I don't know if this is standard terminology, but I find the use of 'headwall' confusing, especially in the abstract, which should be readable without looking at figures. From what I can tell, 'headwall erosion' is basically all primary erosion that is not subglacial. In particular it includes erosion from valley sides, if it does not originate from moraines. To me, headwalls are the mountain sides at cirques at the very top of an accumulation area.

p.1, l.14/15: rewrite sentence or split into two

p.2, l.1: I'm not a fan of the proliferation of acronyms. You only use debris-covered glaciers two or three times, just write it out.

p.2, l.1: The numbers are not clear: Is 11% the area that is debris covered, or are 11% of the glaciers debris-covered glaciers? Also, you must mean 'ice area' not 'ice mass'

p.2, l.15-: See my overall comment 1.

p.2, l.26: are -> is; vary -> varies

p.3, l.4/5: Why are you mentioning this here? What is the relevance of englacial vs supraglacial transport?

p.3, l.8: larger -> longer

p.3, l.1-16: see my comment 1.

p.4, l.15/16: It is not clear to me what you're doing here. Is it legitimate to remove outliers, because they might represent 'extreme events'? A large event should not be ignored if it is not just a data artifact? I suppose you have a way of checking this; the imagery should show large events.

sec. 3.3: When using correlation techniques on debris cover it seems you have to be careful with processes that can generate apparent motion, such as back-wasting of cliffs? The flow field in Fig. 6 looks very odd with vectors pointing across the glacier.

p.4, l.23/24: Can you be more specific how you capture large events that happen 'within a few minutes' and how this information is used?

sec 3.6: Isn't debris that falls onto the glacier in the accumulation area and then transported englacially protected from rounding and will thus preserve its angularity?

p.6, l.20: You say most change occurs on the loose part (0.35 m/yr), but this is almost the same rate as the average (0.31 m/yr) mentioned two lines earlier.

p.6, l.21/22: Again, these rates are not comparable. One is a long term denudation rate, lateral moraine erosion cannot proceed at such high values for centuries; the sediment will be exhausted relatively quickly.

sec. 4.3: I suggest rewriting the first few sentences, because the same things are presented twice, first as averages and then as ranges. This is confusing.

p.8, l.20: What is 'advection of a prominent zone'?

p.8, l.24/25: I don't understand this sentence

p.8, l.27: '... of this process.' What process? The whole paragraph should be edited for clarity.

p.8, l.30: Is it legitimate to assume something to be small, just because it's infrequent? For example, individual large landslides might only occur on decadal timescales or longer, but can dominate the sediment budget of a glacier.

p.9, sec. 4.5: Significant figures! Don't state numbers to sub-meter, which is meaningless in this context.

p.9, l.6: what is the 'observed runout length'? Is that extracted from imagery?

p.9, l.26: check sentence for missing word

p.9, l.29: why 'through the ice'? Isn't it just being dragged along the sides?

p.10, l.7: delete 'being'

p.11, l.1: I would hope that observations are plausible! The question is whether the interpretation of the observations are plausible.

p.11, l.1: an -> a

Figure 1: It took a bit of reading and re-reading to understand the difference between process b and c2; this comes back to the issue of what a headwall is

Figure 2 would be nice if it could be made bigger

Figure 6: give a scale for the velocity arrows.

Martin Truffer
* * *

---

## Referee Comment (RC3) · M.P. Kirkbride (Referee) · 9 Nov 2018

Teun van Woerkom's paper is about a specific focus of geomorphological activity in the deglaciating landscape, and as such is of significant interest. The quality of the data is very high and demonstrates very well patterns of surface change in localities bordering a wasting debris-covered glacier tongue, where interactions between glacier thinning erosion of steep proximal moraine faces, and transfer of debris from moraine to glacier are very active. Unfortunately, from a geomorphological perspective I find the explanations of the patterns identified to be confusingly written. It's hard for a first-time reader to figure out what the authors' understanding of process action actually is, and

ultimately I'm afraid I cannot agree with their calculations and interpretations

There seem to be two reasons for this. First, the terminology as confusing, and some basic geomorphological terms are used loosely to give a false impression of what is meant. "Erosion" is particularly used misleadingly, where measured surface lowering seems to be assumed to equal "erosion" (sensu removal of debris by a transporting medium).

Second, I think there is a problem with the research design. It seems the authors have collected their high-resolution topographic data, calculated the DEM difference maps, and then attempted to interpret them on the basis of elevation change. There is no geomorphological assessment of the sites on the ground. This results in some strange and probably incorrect interpretations which even quite short field visits would have corrected. Better to observe details in the field, map the features, then use that as a basis for carrying out the topographic analysis which then quantifies processes which have already been identified and interpreted empirically, and not by inference from remotely-sensed data. My detailed comments below give examples of this problem.

Detailed comments

Page/line

1/3. Replace "or" with "and" 2/3. Use the standard term for this thickness: "critical thickness". Perhaps acknowledge that this value is actually a variable, and what Østrem found in Sweden may not apply to a Nepalese glacier.

2/16. Surely the headwall extent is a key control as well.

2/32. Here and elsewhere, frost action is hardly mentioned as a contributor to detachment of particles from moraine slopes. Lateral moraines are typically silty and frost-susceptible, and in winter and spring ice crystal growth detaches a large amount of debris which wind and wash then remove.

3/5. Make it clear from the outset that all rates are rates of vertical lowering, and not of

horizontal retreat, both in this study and others cited (notably Curry et al. 2006).

3/19. Just an observation, but this latitude/longitude location does not plot anywhere near Lirung Glacier in Google Earth. I am not saying this location is incorrect, but I wonder where the error lies?

Generally: there are no data on the gradients of the eroding proximal faces of the lateral moraines. This is important data, because for the same rate of vertical lowering, a steeper moraine face will be releasing less debris than a gentler moraine face. A quick calculation shows that for a vertical lowering of 0.31 m/yr (p.1 l.8), a 55 degree face will retreat horizontally by 0.22 m/yr but a 65 degree face by only 0.14 m/yr. If the two faces are similar height, the gentler one will release much more debris in a year. The values of vertical lowering are not therefore very comparable between different locations without some gradient data: a conversion to horizontal retreat rates of the steep moraine faces would be useful.

5/9. "Washout" zone is introducing a new term, and I question whether this is necessary because standard geomorphological terms exist already for debris slopes below retreating faces. If the dominant process is rockfall, it's a talus. If debris flow, it's a colluvial apron or cone or simply slope. If dominant process is uncertain of complex, use a simple morphological term such as debris apron. (Also p.8 l.1 and other places where "loose part of the moraine" is used: is this the same zone?)

6/13. "decrease", not "decreases".

6/19 "Most elevation change occurs in the lower loose part of the moraine...". The terminology and interpretation of this "lower loose part" of the moraine is where I think there are the main problems with the paper. What follows is full of confusing explanation and misleading interpretations. Having said this, all could be easily corrected with some thought and circumspection about the wider setting of the study sites. What I mean will become clear in the comments below.

6/19 – 6/20 et seq. A key finding is that the "lower loose part of the moraine" (presumably equivalent to the "washout zone") has lowered more quickly than the upper eroding face. This lowering is variously described as "erosion" of the slope (p.7 l.24) or "slumping" (p.7 l.16). The first of these is strange given that this is a depositional slope, and it should be aggrading. The second will be discussed in due course. A further curious observation is "a patch of higher erosion... below the firm zone" which the authors explain as wash "plunging" off the steep slope above (see below). This whole set of interpretations is riddled with problems which I will discuss in the line-by-line points below.

6/20-22. I find this comparison of lowering rates confusing. It needs to be more clearly stated what is being compared with what.

6/23-24. These differences could be the result of slope angle differences (see above).

6/28. Delete "m yr-1" after "0.02".

7/3. "slumps and rockfall" ?

7/6 – 7. If I am correct in what this sentence is referring to (and I'm uncertain about this after looking at Figure 5A), I wonder whether this refers to a short steep step of fresh-looking till which commonly forms the very base of the eroding till cliff, and right at the very top of the debris apron below. This is a very common feature of lateral moraines above thinning debris-covered ice generally. Though I haven't read about it in the literature, it has always seemed to me that it forms by lowering of the debris-covered ice at the base of the steep moraine wall: nothing more complicated than that. This paper actually seems to demonstrate this, because the debris aprons are lowering and not aggrading upwards. This just creates a short slope segment marking (probably) that melt season's lowering and separation of the debris slope below from the steep moraine face (the "firm zone"?) above. It's nothing to do with greater erosion by water plunging off the wall above: it's too continuous and uniform to be caused by this.

All the evidence presented strongly suggests that the debris apron isn't eroding, but being gently let down by ablation of glacier ice underneath. Whether this ice is still connected to the glacier or not is secondary: the important point is that its surface is lowering. (This, after all, is what exposes the proximal moraine faces in the first place).

7/13. Replace "tumbling rocks" with "rock fall". Oversteepening of the slope isn't by water flow processes: it is by downward extension of the slope base as the ice below the slope thins. 7/14-15. "The slump toe….the event": meaning unclear. What slump? What do you mean by "the moraine"? After what "event? Confusing. "The lower part of the moraine….": are you including the debris apron ("washout zone/ lower loose part") as part of the moraine? If so, why? It isn't part of the moraine once the debris has been removed from the face and redeposited below.

7/17-18. Average velocities of what? What process is faster than creep? Creep of what? Why make this comparison? Confusing.

7/20. Permafrost requires seasonally-frozen ground: it doesn't require permafrost. On p8 l.2 you show it's probably not solifluction anyway, because motion is greatest in the wet summer season, so it's slow slump or creep of unfrozen saturated debris.

7/24. "erosion rates" of the "loose part of the moraine"? Is this not surface lowering due to a melting ice core, with slow downslope movement of the debris cover over the ice beneath? In other words, it's behaving like a wasting ice-cored moraine. It cannot be called "erosion" because no debris is being entrained and removed by an external medium. Use of this term is misleading and confusing. It's slow gravitational transport.

8/6. Freeze-thaw cycles are important for causing needle-ice growth and detachment of particles in moraine faces. A problem is that few scientists observe moraine faces in winter and spring, when ice crystal growth is evident and thaw-saturation of silty till gives it a very different consistency to the indurated, dry material we see in dry summer weather.

8/9-12. Again your "erosion" values are slope-angle dependent. They are a lowering value, not erosion rates. As discussed above, the lower slope isn't actually eroding at all, so this comparison is spurious.

8/16. Is the "lower moraine" the same as the washout zone, loose part, etc? There needs to be clarity and consistency of terminology throughout the paper. A schematic diagram showing a cross-section through the moraine and toe area with the different zones labelled would be helpful.

8/16-17. Again, it is being assumed that lowering of the lower gentler depositional slope is by erosion, but no geomorphological evidence of erosional processes is presented: it's all based on the lowering rates. Lowering will almost certainly be controlled by melting buried ice: I would venture that there is little or no erosion generally on this slope segment.

8/17 et seq. At last there's a more realistic explanation being presented related to ice surface lowering letting down the debris which has accumulated on its surface. This section could be written with more conciseness and clarity.

8/31. Why "flow velocity". Flow of what?

8/33. Again, "erosion in the lower part" is mentioned. Lowering does not mean erosion at this site. Geomorphologically, it is a depositional slope unit.

Section 4.5. A much better discussion. The key finding is that erosion of lateral moraine faces adds supraglacial material to marginal strips of the thinning glacier surface, but that this material doesn't reach the central zone. Thus, it cannot be the rate-controlling process influencing the overall rate of thinning of the glacier tongue because ice in the centre is able to melt at a higher rate (though retarded by debris advected from much further upstream). The significance of lateral moraine supply is that it generates lateral ice-cored moraines during the late stage of glacier decay. Therefore the wider implications in Section 4.7 (p.11 l.1-6) are an overstatement of the importance of the

process.

The exception is when glacier confluences form medial moraines from lateral moraines-in-transport (sensu Boulton 1978), which then spread debris across the centre of the ablation zone by secondary dispersal (sensu Kirkbride & Deline 2013) to form full-width debris covers. By this mechanism complete debris covers can form. Perhaps this is worth a mention in the wider implications: but the debris needs to be introduced to the glacier centre some distance upstream.

9/15 "approximately"

Section 4.6 Clast Analysis. This is all fine, but how did you assess the roundness of rounded moraine clasts which have shattered on impact when falling from the moraine, to give some angular edges? Did you measure the sharpest edge or the most rounded?

10/11-14. This calculation should be omitted because lateral moraine supply is clearly demonstrated not to cover the full glacier width, so it is a pointless calculation. Also, your average rate of 0.31 m yr-1 isn't valid because it includes the debris apron below the moraine (assuming I have read p.6 l.16-20 correctly), which I have argued earlier is a depositional slope whose lowering is not due to erosion.

10/14. An annual debris thickness increase of 0.29 m yr-1 would generate a layer 10 m thick in about 35 years. So do you see debris layers this thick over the margins of the glacier? I very much doubt it. This reinforces that your rate is not based on a valid calculation. Yet (l.16) you persist in arguing that it is correct.

10/20. Point (b) is simply not a realistic process. Debris accumulating below most of the lateral moraines is not affected by terminus behaviour.

10/27. Point (c) ditto. This whole set of justifications is spurious.

11/1-6. Overstates the significance of lateral moraine near a terminus. To effectively create a complete debris cover which will affect glacier mass balance, debris needs to be introduced to the transport system much higher upstream. Debris introduced from

lateral sources close to the terminus is of more geomorphological than glaciological interest, by creating ice-cored landforms along the base of lateral moraines (which are increasingly common as glaciers retreat). It is most significant in the later stages of the decay of glacier tongues, so a comparison with models of active debris-covered glaciers (p.11 l.4) is misleading.

Overall, my view (reluctantly) is that the paper needs a very careful rethink and rewrite to possibly then be acceptable for publication.

---

## Referee Comment (RC4) · Anonymous Referee #4 · 20 Nov 2018

General comments:

The hypothesis stated on page 3 of the manuscript proposes that lateral moraines can play an important role in supplying the glacier tongue with debris, especially for glaciers that are stagnating and the tongue is disconnected from the headwall. The goals of the paper are to quantify erosion rates across a 5-year period by using high-resolution DEMs. The dataset provided by this paper is remarkable, very high resolution and from a remote location whose future demise will wreak havoc on downstream communities. This large, alpine glacier undoubtedly warrants current research and the negative feed-back cycle created by debris insulating the glacier tongue is an understudied process.

Overall, I find that the study is both overly complex and overly simplified at the same time. To improve the manuscript, the authors could remove some datasets, such as the clast shape analysis, as the population of clasts sampled seems far too small to gain any interpretable insights. This would remove the field-based component of the manuscript, but I don't think that would weaken the paper. On the contrary, I think the paper could be strengthened by using only the remotely sensed data. The authors present many erosion rate calculations for different sections of the moraine, but they never use those data to predict how long (in years, e.g.) it would take for the glacier tongue to remove a certain amount of debris. Although not part of the manuscript in its current state, I think the manuscript could be improved it the authors added a predictive component to their research, like answers questions such as: if the debris cover is adding a negative feedback to the glacier ice, will the debris allow the ice to persist past the year that the glacier ice is expected to melt? And if so, by how many years? At what time point would the lateral moraines deflate enough so that they are not contributing debris to the glacier tongue?

Apart from the technical issues I present below, I find that the authors use the term 'erosion' in a misleading way, and could replace it with 'transport' in some situations. Erosion implies that the material is carried away from the glacier system, as opposed to being transported to lower on the glacier tongue, thus leading to the well-insulated glacier snout that they discuss. If the authors have a mechanism to prove that the material is eroded than they can continue using that term, otherwise it could be replaced with transport.

Specific comments:

As for the writing of the manuscript, I think lines 1-16 on p. 3 can be written so that a testable hypothesis is proposed. The authors list approximate erosion rates from the Alps and from Norway, which could be higher due to the contribution of headwall erosion. The authors need to articulate what they are testing, instead of stating their goals. Perhaps they could add "by quantifying erosion rates of lateral moraines from a

ESurfD
DCG in Nepal, we will test the hypothesis that glaciers with disconnected tongues will have lower erosion rates than those with tongues connected to the headwalls". Lines 12-16 on page 3 do a good job of explaining the goals of the paper, but it is lacking a testable hypothesis.

Technical corrections:

Page 2, Line 31: The phrase "gullied upper part" is vague, I would suggest changing it to more descriptive language. I don't get a sense of what part of the moraines are gullied.

Page 2, Lines 34-35: The sentence: "The transport processes on the moraines are most active directly after deglaciation" makes intuitive sense but is a large claim. Does this imply that your modern study falls within the period of most active transport? I would either add another sentence or two linking this claim to your study, or remove it all together.

Page 4 line 10, that is a big assumption that no melting core exists in the moraine. Do they have any data that would substantiate this claim?

Page 6, Lines 12-15: This sentence reads as a hypothesis since the authors are predicting what the . It reads as more of a topic sentence and could be moved to the beginning of the paragraph.

Page 6, Lines 19-20: Here the terms "lower loose" and "upper firm" start getting usage as descriptors for parts of the moraine. I understand that terms loose and firm correlate to the different erosion rates, but perhaps more descriptive terms could be used, such as "low erosion area" and "higher erosion area".

Page 7, Lines 19-20: Solifluction is mentioned as a main transport mechanism, yet little discussion in the text is allocated to this process. Is there a way the authors can interpret solifluction processes from their DEM data?

Page 11, Line 13: Change mechanisms to mechanism

Page 11, Line 22: Change are to is

Figures 4 and 5: It would be useful to have an index map of where these images/data are located on the glacier.

---

## Author Comment (AC1) · 30 Jan 2019

See attachment

Please also note the supplement to this comment:
https://www.earth-surf-dynam-discuss.net/esurf-2018-63/esurf-2018-63-AC1-supplement.pdf

––––––––––––––––––––––––––––

---

## Author Comment (AC2) · 30 Jan 2019

**Response to RC1**

Dear Martin Truffer,

Thank you for your comments, which indicated some important points which we address below. First we respond to the two major issues raised, afterwards the detailed comments are addressed individually. We hope that you agree with the changes we propose based on the suggestions.

**1) In the paper erosion from several sources (headwall, lateral, basal) is treated similarly and independently. But this is not really the case. All sediment in lateral moraine has originated from headwall erosion, transported by the glacier and is now remobilized. It's not primary erosion. So it doesn't really make sense to compare rates, these are two different things. It would improve the paper greatly if this distinction was clearly made in the Introduction as well as the Discussion. This would also outline some of the issues that need to be addressed in coupled debris/ice flow models, namely sediment deposition and re-activation.**

We agree that this is an important distinction, which should be made clear. Therefore this direct comparison is removed from the discussion and we have specifically referred to it as a 'remobilization' therein (P2 (L16)), pointed out the lack of this process in relevant models (P12 (L8-11)) and have also reformulated in the Introduction (P2 (L29)). We have kept the comparison to rates found in (Watanabe et al., 1998) in the introduction, as their rates also include remobilized lateral moraine erosion and that is probably the reason they found much higher elevation change rates as generally observed on rock walls. For the rates found in other studies, the lateral moraine has the same characteristics as those observed in this paper, though not anymore bordered by a glacier. Therefore we think that these rates are comparable.

**2) The derived rates are based on the assumption that none of the elevation changes on the moraine are due to melting ice. This seems like a big assumption and it is only mentioned in one sentence and no potential errors due to this assumption are discussed.**

As noted by several reviewers, we have failed to address the presence of an ice-core within the moraine in our original manuscript. While there is no actual evidence of an ice-core, the formation process of lateral moraines suggests they are likely present and field evidence from other debris-covered glaciers in HMA exists (e.g. (Hambrey et al., 2009)). However at least for the case of Lirung and other debris-covered tongues in the Langtang catchment, field observations suggests that these ice cores are covered in a very thick mantle of debris, very likely larger than 2 m, contrary to more thinly covered moraines elsewhere (e.g. (Lukas et al., 2005)). During many walks in multiple field seasons on the glacier tongue, including the flanks, nowhere along or within 50 m of the foot of the moraine have we observed any ice, neither as ice cliffs nor as ice covered under thin moraine material. Furthermore, none of the debris thickness measurements ever taken in the field close to the moraine on the glacier surface where thinner than 50 cm (see e.g. (Ragettli et al., 2015)), and since the debris progressively thickens (see e.g. (Nicholson et al., 2018) towards the moraine it is likely much thicker there (Nicholson et al., 2018). We estimate the maximum downwasting rate under the moraine by quantifying the elevation change in a relative flat zone of 20 meter wide close to the moraine. We removed sections which showed clear depositional features, to limit the noise caused by debris deposition. The maximum downwasting rate is approximately 0.6 m yr$^{-1}$. The top melt of buried ice declines exponentially with increasing debris-cover thickness (Östrem, 1959; Schomacker, 2008), but

as debris thickness on the glacier is generally > 50 cm (McCarthy et al., 2017; Ragettli et al., 2015), the decline of melt rates underneath the moraine due to the additional debris thickness is relatively small.

Assuming a maximum downwasting rate of 0.6 m yr$^{-1}$ due to the ice core, the remaining elevation change due to mass transport in this part is +0.19 m yr$^{-1}$. This results in a significantly reduced rate of material reaching the glacier surface and hence our interpretations of the results. As a consequence, we removed the explanation on page 10 (line 20-27) as suggested.

While we acknowledge that further detailed analysis could be carried out to ascertain presence of a potential ice core (i.e. GPR) or to understand the processes in recent decades and centuries, our aim here was to determine the volume of debris that moved onto the glacier surface in recent years using an UAV. We believe that the use of high-resolution DEMs in combination with geomorphological analysis has great potential to understand the dynamics of debris-covered glacier tongues.

**Detailed comments:**
**p.1, l.5: I don't know if this is standard terminology, but I find the use of 'headwall' confusing, especially in the abstract, which should be readable without looking at figures. From what I can tell, 'headwall erosion' is basically all primary erosion that is not subglacial. In particular it includes erosion from valley sides, if it does not originate from moraines. To me, headwalls are the mountain sides at cirques at the very top of an accumulation area.**

Indeed the term headwall gives a wrong impression, as it is mostly used for steep scarps (or cirques) at the onset of erosion or accumulation area. The term is therefore changed to rockwall throughout the manuscript, which is the general terminology for steep bedrock faces bordering glaciers.

**p.1, l.14/15: rewrite sentence or split into two**
Changed

**p.2, l.1: I'm not a fan of the proliferation of acronyms. You only use debris-covered glaciers two or three times, just write it out.**
Changed throughout the manuscript

**p.2, l.1: The numbers are not clear: Is 11% the area that is debris covered, or are 11% of the glaciers debris-covered glaciers? Also, you must mean 'ice area' not 'ice mass'**
We meant that that 11% of all glacier areas are debris-covered. Secondly, we do indeed refer to ice mass (Kraaijenbrink et al., 2017; p.2, l.2/3). This has been modified.

**p.2, l.15-: See my overall comment 1.**
We have added the distinction between direct and indirect (remobilized) sources of debris.

**p.2, l.26: are -> is; vary -> varies**
Changed

**p.3, l.4/5: Why are you mentioning this here? What is the relevance of englacial vs supraglacial transport?**
This was mentioned to indicate that debris is transported over the surface and adds to the debris cover. Sentence changed.

**p.3, l.8: larger -> longer**
Changed

**p.3, l.1-16: see my comment 1.**
Rates can indeed not always be compared directly. Important differences in source areas or study sites are now mentioned more clearly.

**p.4, l.15/16: It is not clear to me what you're doing here. Is it legitimate to remove outliers, because they might represent 'extreme events'? A large event should not be ignored if it is not just a data artifact? I suppose you have a way of checking this; the imagery should show large events.**
It is definitely not legitimate to remove extreme events from the data, and such is also not the goal of the 10-90 percentile range. Our text was misleading in this regard. It may be true that some signal is also removed using this method, but the imagery still shows larger events as rockfall and intense gully erosion, which we managed to retain. We have adapted the text accordingly in P4 (L19-21).

**sec. 3.3: When using correlation techniques on debris cover it seems you have to be careful with processes that can generate apparent motion, such as back-wasting of cliffs? The flow field in Fig. 6 looks very odd with vectors pointing across the glacier.**
Indeed, processes that cause apparent motion do affect the surface velocities computed by (Kraaijenbrink et al., 2016). However we can account for this problem by choosing the window size of the cross-correlation appropriately and validating the results with orthophotos. As can be seen from Fig. 6, ice cliffs are not present where the vectors point inwards, and no process that generates apparent (but not actual) motion seems to play a role here. The pattern of flow velocity is furthermore recurring in multiple years (Kraaijenbrink, in prep.) and can be explained with flow (Kraaijenbrink et al., 2016). Therefore it is likely that this velocity actually indicates the surface velocity. We updated the caption to further elaborate on the origin of surface velocities.

**p.4, l.23/24: Can you be more specific how you capture large events that happen 'within a few minutes' and how this information is used?**
Unfortunately our wording here was confusing. Large events, where debris flows or rockfalls travel over multiple window sizes between the two time steps (the maximum window size used is 26.5 m), are not captured as velocity but rather as noise in the algorithm and hence do not influence the velocity fields. While we still capture such events visually and in the DEM differencing, they do not bias our average velocities on the moraines. We have changed the text respectively on P5 (L1-2).

**sec 3.6: Isn't debris that falls onto the glacier in the accumulation area and then transported englacially protected from rounding and will thus preserve its angularity?**
We distinguish between passive and active glacial transport. Passive transport consists of most englacial transport and material at the glaciers surface, on which no large forces are exerted. Active glacial transport takes place at the glacier bed or within deforming subglacial sediment (while forming the lateral moraine in times of glacial advancement in this case), and does cause the angularity to decrease debris under high pressures (Benn & Ballantyne, 1994). In the case of passive transport, the original angularity of the clasts is approximately maintained (Benn & Ballantyne, 1994). Therefore we assume that material on the glacier, which originated from the rockwalls in the accumulation area (and might afterwards be englacially transported), is more angular than the clasts currently on the moraines. We adapted sec 3.6 to make this distinction more clear.

**p.6, l.20: You say most change occurs on the loose part (0.35 m/yr), but this is almost the same rate as the average (0.31 m/yr) mentioned two lines earlier.**
The 0.35 m yr$^{-1}$ for the loose part of the moraine was a typo throughout the manuscript. The correct value (0.41 m yr$^{-1}$) could already be found in Table 3. The remaining difference is caused by differences in area, as the loose area covers 74% of the total moraine area, and the firm part only covers 26%.

**p.6, l.21/22: Again, these rates are not comparable. One is a long term denudation rate, lateral moraine erosion cannot proceed at such high values for centuries; the sediment will be exhausted relatively quickly.**
That is correct and the comparison is removed from the manuscript.

**sec. 4.3: I suggest rewriting the first few sentences, because the same things are presented twice, first as averages and then as ranges. This is confusing.**
We combined the presentation of both elevation change values and ranges at the start of section 4.3.

**p.8, l.20: What is 'advection of a prominent zone'?**
We have adapted the terminology to 'dispersal of the deposited slump sediments', which we hope will make this clear.

**p.8, l.24/25: I don't understand this sentence**
Changed paragraph in line with next comment.

**p.8, l.27: '... of this process.' What process? The whole paragraph should be edited for clarity.**
Changed

**p.8, l.30: Is it legitimate to assume something to be small, just because it's infrequent? For example, individual large landslides might only occur on decadal timescales or longer, but can dominate the sediment budget of a glacier.**
We agree that infrequent events can be the main source of sediment on a supraglacial surface, and it is not unlikely that this is also the case for Lirung Glacier, where large mass movement events are observed near the rockwalls. We however argue that the moraine-derived sediment budget is not dependent on infrequent events for three reasons:
-   Really large events are not a possibility, as the source area is small. The steep part of the moraine is generally only 30-40 meters wide, with a maximum of 60. With a high gully density, the source area per gully is really small and prevents the occurrence of large flows. Furthermore such events were also not observed while we revisited the site biannually between 2013 and 2018.
-   Individual tumbling rocks are observed almost each time step. However, with only 1-2 rocks per half year, the removed volume is in the order of 100 $m^3$ $yr^{-1}$. This can be neglected in respect to the total volume of 126000 $m^3$ $yr^{-1}$.
-   The same goes for 'larger' landslides, which are only observed once in the studied period. Its volume of 2000 $m^3$ is still small in regard to the total volume. We added our reasoning to the manuscript in P9 (L19-22).

**p.9, sec. 4.5: Significant figures! Don't state numbers to sub-meter, which is meaningless in this context.**
Changed

**p.9, l.6: what is the 'observed runout length'? Is that extracted from imagery?**
The maximum observed runout length is indeed manually extracted from imagery, where the active depositional area can often be distinguished from the inactive subglacial debris by differences in roughness and color. For example in the figure below, debris flow runout lobes are recognized and outlined.

[Figure]

**p.9, l.26: check sentence for missing word**
Word added.

**p.9, l.29: why 'through the ice'? Isn't it just being dragged along the sides?**
This was indeed misleading and should be 'dragged along the glacier bed'. We have changed this in the text at P10 (L29-30).

**p.10, l.7: delete 'being'**
Deleted.

**p.11, l.1: I would hope that observations are plausible! The question is whether the interpretation of the observations are plausible.**
We removed this part of the sentence, and focused on the interpretation.

**p.11, l.1: an -> a**
Changed

**Figure 1: It took a bit of reading and re-reading to understand the difference between process b and c2; this comes back to the issue of what a headwall is**
The terminology of both b and c2 has been adapted. Rockwall is now used to indicate steep bedrock faces bordering the glacier, and rockfall is used to refer to both tumbling rocks and rockfall of the steep bedrock faces, as the mass movement process is the same.

**Figure 2 would be nice if it could be made bigger**
Figure is enlarged a bit. To further explain the study area characteristics, an additional figure with cross-glacier transects is created, and old figure 3 (now 4) is enlarged too.

**Figure 6: give a scale for the velocity arrows.**
The velocity arrows were non-linearly scaled to best suit visual interpretation. If linearly scaled, most arrows would become small and illegible, most definitely the ones on the moraines. Therefor the legend label is now changed to Surface flow direction, to imply that magnitudes are just indicative.

**References**

Benn, D. I., & Ballantyne, C. K. (1994). Reconstructing the transport history of glacigenic sediments: a new approach based on the co-variance of clast form indices. *Sedimentary Geology*, *91*(1–4),

215–227. https://doi.org/10.1016/0037-0738(94)90130-9

Hambrey, M. J., Quincey, D. J., Glasser, N. F., Reynolds, J. M., Richardson, S. J., & Clemmens, S. (2009, December). Sedimentological, geomorphological and dynamic context of debris-mantled glaciers, Mount Everest (Sagarmatha) region, Nepal. *Quaternary Science Reviews*. Elsevier Ltd. https://doi.org/10.1016/j.quascirev.2009.04.009

Kraaijenbrink, P., Meijer, S. W., Shea, J. M., Pellicciotti, F., De Jong, S. M., & Immerzeel, W. W. (2016). Seasonal surface velocities of a Himalayan glacier derived by automated correlation of unmanned aerial vehicle imagery. *Annals of Glaciology*, *57*(71), 103–113. https://doi.org/10.3189/2016AoG71A072

Lukas, S., Nicholson, L. I., Ross, F. H., & Humlum, O. (2005). Formation, meltout processes and landscape alteration of high-arctic ice-cored moraines - Examples from Nordenskiöld land, central Spitsbergen. *Polar Geography*, *29*(3), 157–187. https://doi.org/10.1080/789610198

McCarthy, M., Pritchard, H., Willis, I., & King, E. (2017). Ground-penetrating radar measurements of debris thickness on Lirung Glacier, Nepal. *Journal of Glaciology*, *63*(239), 543–555. https://doi.org/10.1017/jog.2017.18

Nicholson, L. I., McCarthy, M., Pritchard, H., & Willis, I. (2018). Supraglacial debris thickness variability: Impact on ablation and relation to terrain properties. *The Cryosphere Discussions*, *12*(12), 1–30. https://doi.org/10.5194/tc-2018-83

Östrem, G. (1959). Ice Melting under a Thin Layer of Moraine, and the Existence of Ice Cores in Moraine Ridges. *Geografiska Annaler*, *41*(4), 228–230. https://doi.org/10.1080/20014422.1959.11907953

Ragettli, S., Pellicciotti, F., Immerzeel, W. W., Miles, E. S., Petersen, L., Heynen, M., et al. (2015). Unraveling the hydrology of a Himalayan catchment through integration of high resolution in situ data and remote sensing with an advanced simulation model. *Advances in Water Resources*, *78*, 94–111. https://doi.org/10.1016/j.advwatres.2015.01.013

Schomacker, A. (2008). What controls dead-ice melting under different climate conditions? A discussion. *Earth-Science Reviews*, *90*(3–4), 103–113. https://doi.org/10.1016/j.earscirev.2008.08.003

Watanabe, T., Dali, L., & Shiraiwa, T. (1998). Slope denudation and the supply of debris to cones in Langtang Himal, Central Nepal Himalaya. *Geomorphology*, *26*(1–3), 185–197. https://doi.org/10.1016/S0169-555X(98)00058-0

---

## Author Comment (AC3) · 30 Jan 2019

**Response to RC2**

Dear Simon Pendleton,

Thank you for your constructive comments, which we address in detail below. First we respond to the specific issues raised, afterwards the detailed comments are addressed individually.

**Overall Comments: I applaud the authors for tackling a complex problem that is nonetheless very pertinent for understanding the dynamics of debris covered glaciers. Using higher resolution DEMs to investigate many aspects of moraine evolution through space and time is an important task. However, I have a few concerns that I believe should be addressed before this manuscript moves forward towards publication. First, the assumption of no ice-cored moraines is substantial, and one that should be backed-up with field observations or other evidence. Debris covered glaciers are notorious for producing ice cored moraines, and elevation changes triggered by meltout could be an important component.**

As noted by several reviewers, we have failed to address the presence of an ice-core within the moraine in our original manuscript. While there is no actual evidence of an ice-core, the formation process of lateral moraines suggests they are likely present and field evidence from other debris-covered glaciers in HMA exists (e.g. (Hambrey et al., 2009)). However at least for the case of Lirung and other debris-covered tongues in the Langtang catchment, field observations suggests that these ice cores are covered in a very thick mantle of debris, very likely larger than 2 m, contrary to more thinly covered moraines elsewhere (e.g. (Lukas et al., 2005)). During many walks in multiple field seasons on the glacier tongue, including the flanks, nowhere along or within 50 m of the foot of the moraine have we observed any ice, neither as ice cliffs nor as ice covered under thin moraine material. Furthermore, none of the debris thickness measurements ever taken in the field close to the moraine on the glacier surface where thinner than 50 cm (see e.g. (Ragettli et al., 2015)), and since the debris progressively thickens (see e.g. (Nicholson et al., 2018) towards the moraine it is likely much thicker there (Nicholson et al., 2018). We estimate the maximum downwasting rate under the moraine by quantifying the elevation change in a relative flat zone of 20 meter wide close to the moraine. We removed sections which showed clear depositional features, to limit the noise caused by debris deposition. The maximum downwasting rate is approximately 0.6 m $yr^{-1}$. The top melt of buried ice declines exponentially with increasing debris-cover thickness (Östrem, 1959; Schomacker, 2008), but as debris thickness on the glacier is generally > 50 cm (McCarthy et al., 2017; Ragettli et al., 2015), the decline of melt rates underneath the moraine due to the additional debris thickness is relatively small.

Assuming a maximum downwasting rate of 0.6 m $yr^{-1}$ due to the ice core, the remaining elevation change due to mass transport in this part is +0.19 m $yr^{-1}$. This results in a significantly reduced rate of material reaching the glacier surface and hence our interpretations of the results. As a consequence, we removed the explanation on page 10 (line 20-27) as suggested.

While we acknowledge that further detailed analysis could be carried out to ascertain presence of a potential ice core (i.e. GPR) or to understand the processes in recent decades and centuries, our aim here was to determine the volume of debris that moved onto the glacier surface in recent years using an UAV. We believe that the use of high-resolution DEMs in combination with geomorphological analysis has great potential to understand the dynamics of debris-covered glacier tongues.

**Also, the large variability of reported elevation change measurements and the overlap between measurements makes delineating significant differences between measurements difficult (see Fig. 1 attached). A more detailed treatment of variability in reported elevation changes would greatly benefit the manuscript.**

We have addressed your concerns relating to the variability below.

**Specific Comments: The Authors allude (in the abstract in on pg. 3, ln 2) to the fact that debris input from lateral moraines is more important for retreating glaciers or for glaciers with stagnant tongues. What is not explicitly stated is that contribution from lateral moraines only occurs when the surface of the glacier is below the crest of the lateral moraines; though this is partly common sense, I think that explicitly stating this in the introduction is worthwhile. It may also be pertinent to expand upon the temporal variability with regards to debris input; an advancing glaciers will have a higher proportion of debris sourced from the headwall, while a retreating glacier can accept debris input from lateral moraines. The authors speak to a this later in the discussion, but setting the stage in the introduction may better lead into the later discussion.**

The introduction now includes a broader description of the study site at P3 (L2-8), including a description of those situations in which lateral moraines can be an important source of debris. We also explicitly state this to be of importance when the glacier downwastes below the crest at P3 /L2-3). The new Figure 3 also illustrates our arguments better. This greatly improves the discussion of debris distribution onto the glacier. We added to the conclusions that the changing contribution of lateral moraine debris to a debris covered glacier in relation to different glacier recession scenarios should be a future research focus.

**Section 2 states that the moraines are 'disconnected' from upper slopes. I take this to mean that the outboard face of the lateral moraines have an opposite aspect as the valley wall (i.e., debris falling from the valley walls will likely not travel over the moraine onto the glacier). Perhaps obvious to persons on the ground observing the glacier system, but explicitly stating this is important for justifying no debris input from valley walls. Also, comparing Fig. 1 to current google earth imagery, it is not clear if the upper part of the mapped moraines are actually disconnected from the valley wall.**

We added additional explanation to the introduction to clarify this. Furthermore cross sections from the glacier are added to a new figure (Figure 3) to make the disconnect more clear.

**After referring to the potential for ice cored moraines potentially contributing mass as they degrade over time in the introduction, the Authors assume that no melting ice core exists in the moraine (such that all elevation changes are due to sediment transport off the moraines; pg. 3, ln 10). In a seemingly contradictory statement, the authors discuss a '. . .hillshade with a hummocky appearance is an indicator for subdebris ice' (pg. 5, ln 4) when discussing the lateral moraines. Ice cored moraines are generally quite commonly associated with debris covered glaciers (Clark et al., 1994), and this catchment is no exception. The authors provide no observations or other evidence to support their assumption that the moraines are not ice cored. Ice cores within moraines are known to persist for thousands of years and can help maintain steeper moraine slopes (Crump et al., 2017), which are present in the Lirung glacier. Visual observations from current Google Earth imagery shows that appears to be a consistent debris cover across most of the glacier and pro-**

**glacial area, which suggests that the glacier very likely could have formed ice cored moraines in the past. Without concrete evidence that the existing lateral moraines are not significantly ice cored, I am hesitate to agree with the assumption that all elevation changes on the moraines are solely due to sediment transport.**

In this case, P5 L13-14, the hummocky appearance is indeed used as an indicator for subdebris ice. However, we use this to construct the lower boundary of the moraine, i.e. when the surface is hummocky  that part is not classified as moraine and excluded from our elevation difference calculation.

As indicated above, we have taken into consideration the ice cored moraine in the revised manuscript. We have now included this throughout the manuscript.

**In section 4.1 a vertical error of 0.02  ± 0.33 m y-1 is quoted, citing (Immerzeel et al., 2014). Upon further reading of (Immerzeel et al., 2014), it is difficult to tell, but it appears that the off-glacier area used to compute this error is actually the lateral moraines themselves, which are likely to change in elevation between May and Oct 2013); perhaps I am interpreting this incorrectly, but moraines don't seem like an ideal location to compare the accuracy of DEMs due to their changing elevation.**

The chosen areas are valid locations to assess the accuracy, as dGPS measurements through time show that there is no trend in elevation difference on the off-moraine terrain. Most of our dGPS points are located well away from the moraine (**Error! Reference source not found.**; top 8 locations on the Western side and top 5 locations on the Eastern side). All others are located on or close to the distal slope and one at the outlet of the terminus lake.

Elevation changes on the distal slope of the moraines, as evidenced by comparing areas on all DEMs as well as comparing the DEMs to the dGPS points in 2015, show differences in the range of approximately -1.0 m to 0.7 m. There is no apparent trend and we attribute this to the DEM error than any actual elevation change (Table 1). Furthermore there are paths as well as sensor setups positioned on this distal slope, that have not changed since we visit the field site in 2012.

Similarly, referring to Figure 6 in the original manuscript, the DEMs show positive as well as negative mean annual change in what we referred to as the 'upper gullied section' in the manuscript. On the other hand, there is a clear line separating it from the lower part, where elevation change is distinctly negative, likely also due to dead ice underneath (as visualized for example in Fig. 11f in (Lukas, Graf, Coray, & Schlüchter, 2012)). This difference indicates the probable lack of ice underneath the distal moraine, and supports the dGPS data in showing the stability of the off-glacier terrain used for error computation.

[Figure]

*Figure 1; Locations of dGPS markers and mean differences (m) between dGPS measurement and all DEMs used in this study. Also all differences and their mean (corresponding to Table 1) are plotted, which do not show a distinct trend.*

|  | Mean (m) | Std (m) |
|---|---|---|
| 18/5/2013 | 0.101 | 0.297 |
| 22/10/2013 | 0.072 | 0.277 |
| 18/10/2015 | -0.061 | 0.210 |
| 30/04/2016 | -0.003 | 0.184 |
| 6/10/2016 | -0.017 | 0.302 |
| 20/04/2017 | 0.089 | 0.160 |
| 19/10/2017 | 0.056 | 0.284 |
| 28/4/2018 | -0.005 | 0.312 |

*Table 1: Mean deviations between all dGPS measurements taken at 18-10-2015 and the respective DEMs. The locations are shown in **Error! Reference source not found.**, as well as the entire dataset variability.*

**Maybe the vegetated areas are more stable, so are appropriate to use to constrain DEM accuracy at different timesteps? Also the quoted accuracy of the DEMs is only calculated for DEMs produced in May and Oct of 2013; what about the accuracy of DEMs produced up through 2018? Further reading of the more recent Kraaijenbrink et al. (2016) indicates that "The bulk of the vertical errors at the tie points are within 50cm, and 75% are even within ~25cm." I am by no means an expert in DEM creation, let alone from high resolution UAV imagery, but if this is the actual uncertainty in elevations between DEMs, then the reported elevations changes in this manuscript (e.g., Table 3) are well within this uncertainty. Therefore, how can you be certain that your extracted measurements are above the vertical uncertainty and noise between your DEMs from different**

times? Perhaps I am not reading Kraaijenbrink et al. (2016) correctly, but at minimum, it would be appropriate to add more detail regarding the vertical errors associated with your DEMs into this study (in addition to citing Immerzeel et al. (2014) and Kraaijenbrink et al. (2016)), since changes in elevation are the key component of this study.

It is indeed very likely that the surface under vegetation is most stable. However, it is the surface elevation of leaves, shrubs and grasses that is observed by the UAV, which is highly variable due to vegetation growth. Therefore it is not suitable.

We agree that more information regarding the vertical accuracy should be given, and an additional table (Table 4) is listing the off-glacier offset for all time periods used. More information regarding these uncertainties are given at P7 (L7-10). The value for 2013 is now -0.04± 0.44 m yr$^{-1}$ instead of the previously published 0.02 ± 0.33 m yr$^{-1}$. This difference is caused by the Gorka 2015 Earthquake, which caused a large lateral replacement. After the event (and thus after Immerzeel et al. (2014)) the pre-earthquake data were reprocessed to take account of this offset and to still be able to compare with the full time series.

From these values it is clear that f the signal that we observed on the moraines our data is much larger than the possible error. Locally errors may be larger, but therefore our paper focusses on mean elevation changes and general patterns.

The Authors mask out vegetation areas from the DEMs to ensure that the moraine elevations changes represent real change. I am confused by this. Mass wasting events and sediment transport within vegetated areas aren't real change? Perhaps I am not interpreting this correctly.

Yes, sediment transport in vegetated areas on the proximal moraine slopes certainly happens, but very little is observed during our field visits. In addition as stated above, incorporating vegetated areas would cause errors, as seasonal vegetation dynamics will result in temporally variable deviations between the UAV DEM and the actual surface elevation.  This is clarified in the manuscript at P4 L15-17).

The Authors also state that they correct the DEMs for off-glacier and off-moraines elevation changes with the assumption that those landscapes should be stable. Again it is not entirely clear why the DEMs must be corrected for these large changes; aren't large sediment transport events possible on 'stable' landscapes (though less common)?

Yes, sediment transport is surely also possible on the landscapes that are assumed to be stable. After both field visits and careful examination of the drone images no sign of significant sediment transport was observed, which leads to the conclusion that these should have a stable elevation over the investigated period of time.

Table 3 presents the elevation changes derived from differencing various DEMs over the study period. I am assuming that the mean the 1sigma uncertainty is presented, but this is not explicitly stated in the table (see technical corrections). When plotted, the mean and uncertainties all overlap well within each other's (see attached Fig. 1). Are these measurements statistically different from each other? Enough to back up the arguments regarding the spatial and temporal patterns presented in the manuscript? It would be nice to see the raw data plotted in a histogram to see the distribution of elevation changes within each different moraine region/time period (e.g., Immerzeel et al. (2014)

**Fig. 6). This would allow the reader to get a better sense for how the calculated elevation differences vary within a region and a nice comment to Fig. 6.**

The caption below Table 3 is changed to clarify the 1-sigma uncertainty. We tested the significance within the designed zones (entire moraine, upper gullied section, lower debris apron) and between all time periods. A column is added to Table 3 with the period numbers of statistically different datasets. Differences per time period over the entire moraine are also always statistically different. The results also indicate that there is a significant difference in elevation change between the upper and lower moraine over the entire period.

**Table 3.** Seasonal elevation change values, furthermore divided in upper and lower moraine. The zonal mean ($\mu$) is reported, as well as the 1-sigma standard deviation ($\sigma$). Precipitation is measured at Kyanjing station in 2013 and Langshisha station in all others seasons. Elevation change values are in m yr$^{-1}$, precipitation values in mm and mm hr$^{-1}$. The significance column indicates the periods from which that specific dataset statistically differs ($p < 0.05$).

| Period | Dataset | Entire moraine $\mu$ | $\sigma$ | sig | Gullied upper part $\mu$ | $\sigma$ | sig | Debris apron, lower part $\mu$ | $\sigma$ | sig | Precipitation cumulative | mean intensity |
|---|---|---|---|---|---|---|---|---|---|---|---|---|
| 1 | 2013 / 05 - 2013 / 10 (wet) | -0.39 | 0.79 | all | -0.28 | 0.77 | all | -0.41 | 0.83 | 2, 4-6 | 697 | 0.90 |
| 2 | 2015 / 10 - 2016 / 04 (dry) | -0.17 | 0.44 | all | -0.07 | 0.43 | all | -0.21 | 0.44 | all | 145 | 0.62 |
| 3 | 2016 / 04 - 2016 / 10 (wet) | -0.34 | 0.57 | all | -0.14 | 0.55 | all | -0.41 | 0.59 | 2, 5-6 | 584 | 0.58 |
| 4 | 2016 / 10 - 2017 / 04 (dry) | -0.36 | 0.37 | all | -0.24 | 0.37 | all | -0.41 | 0.36 | 1-2, 5-6 | 172 | 0.84 |
| 5 | 2017 / 04 - 2017 / 10 (wet) | -0.52 | 0.84 | all | -0.19 | 0.82 | all | -0.60 | 0.82 | all | 541 | 0.58 |
| 6 | 2017 / 10 - 2018 / 04 (dry) | -0.22 | 0.98 | all | -0.12 | 0.98 | all | -0.25 | 1.00 | all | 117 | 0.50 |
| 7 | 2013 / 05 - 2018 / 04 (total) | -0.31 | 0.26 | all | -0.16 | 0.26 | all | -0.41 | 0.21 | 2, 5-6 | 2257 | 0.67 |

The histograms of the entire time period are added to Figure 7.

**Technical Corrections: Please see attached PDF for technical corrections.**

Thank you for the technical corrections. We have implemented all. The remaining comments are addressed below.

**P2l23: Expand on this; I think you are trying to say that the moraines are separated enough from the valley wall that the influence of rockfall/debris input from the valley wall is negligible**

Indeed, this is the case for the entire study area. We added Figure 3 and an additional explanation to clarify this.

**P3l30: Given that a majority of your conclusions are based on elevation differences derived from DEMs, supplying details on DEM creation and quantifying the uncertainty here is important.**

More information on DEM creation is added to the manuscript at P4 (L2-7). Furthermore Table 4 and P7 (L7-10) are added with the uncertainty of each of the timesteps.

**P7l27: I'm not sure just reporting the means here is entirely transparent, most of these mean elevation changes have large uncertainties, which are important to include when reporting the data here.**

We included the uncertainty estimate to the reported mean values.

**P8l9: Report uncertainties along with these values.**

This has been included.

**Figure 1: A larger photograph would help to give the reader a better idea of the overall geomorphology of the glacier system. / Figure 3: Could these be bigger maybe?**

This has been included.

**Figure 5: What is the dashed line? Is it the same as in Fig. 4? All parts of the figure need to be defined.**

This is now explained in the legend.

**Figure 6: Are the blank areas within the data where vegetated areas were masked out? if so, that should be stated in the figure caption.**

The caption was changed accordingly.

**Figure 10: Parts of this figure are also quite small and hard to read, could it be made bigger?**

We enlarged the text in the figure.

**Table 3: -units should accompany all columns**

**-please define μ and σ**

Changed

**-A histogram or frequency diagram of each of these would be useful to look at the distribution of elevation changes in space and time**

The histograms of the entire studied period were added to Figure 7.

**References**

Hambrey, M. J., Quincey, D. J., Glasser, N. F., Reynolds, J. M., Richardson, S. J., & Clemmens, S. (2009, December). Sedimentological, geomorphological and dynamic context of debris-mantled glaciers, Mount Everest (Sagarmatha) region, Nepal. *Quaternary Science Reviews*. Elsevier Ltd. https://doi.org/10.1016/j.quascirev.2009.04.009

Immerzeel, W. W., Kraaijenbrink, P. D. A., Shea, J. M., Shrestha, A. B., Pellicciotti, F., Bierkens, M. F. P., & De Jong, S. M. (2014). High-resolution monitoring of Himalayan glacier dynamics using unmanned aerial vehicles. *Remote Sensing of Environment*, *150*, 93–103. https://doi.org/10.1016/j.rse.2014.04.025

Lukas, S., Graf, A., Coray, S., & Schlüchter, C. (2012). Genesis, stability and preservation potential of large lateral moraines of Alpine valley glaciers - towards a unifying theory based on Findelengletscher, Switzerland. *Quaternary Science Reviews*, *38*, 27–48. https://doi.org/10.1016/j.quascirev.2012.01.022

Lukas, S., Nicholson, L. I., Ross, F. H., & Humlum, O. (2005). Formation, meltout processes and

landscape alteration of high-arctic ice-cored moraines - Examples from Nordenskiöld land, central Spitsbergen. *Polar Geography*, *29*(3), 157–187. https://doi.org/10.1080/789610198

Nicholson, L. I., McCarthy, M., Pritchard, H., & Willis, I. (2018). Supraglacial debris thickness variability: Impact on ablation and relation to terrain properties. *The Cryosphere Discussions*, *12*(12), 1–30. https://doi.org/10.5194/tc-2018-83

Ragettli, S., Pellicciotti, F., Immerzeel, W. W., Miles, E. S., Petersen, L., Heynen, M., … Shrestha, A. (2015). Unraveling the hydrology of a Himalayan catchment through integration of high resolution in situ data and remote sensing with an advanced simulation model. *Advances in Water Resources*, *78*, 94–111. https://doi.org/10.1016/j.advwatres.2015.01.013

---

## Author Comment (AC4) · 30 Jan 2019

Sorry, I accidentally uploaded the wrong document. The correct document can be found under my RC2 reply.

---

## Author Comment (AC5) · 30 Jan 2019

**Response to RC3**

Dear Martin Kirkbride,

Thank you for your substantial comments, which prompted us to rethink certain parts of the analysis and which have improved the manuscript considerably to our opinion. First we respond to the general issues raised, afterwards the detailed comments are addressed point by point.

**Teun van Woerkom's paper is about a specific focus of geomorphological activity in the deglaciating landscape, and as such is of significant interest. The quality of the data is very high and demonstrates very well patterns of surface change in localities bordering a wasting debris-covered glacier tongue, where interactions between glacier thinning erosion of steep proximal moraine faces, and transfer of debris from moraine to glacier are very active. Unfortunately, from a geomorphological perspective I find the explanations of the patterns identified to be confusingly written. It's hard for a first-time reader to figure out what the authors' understanding of process action actually is, and ultimately I'm afraid I cannot agree with their calculations and interpretations**

**There seem to be two reasons for this. First, the terminology as confusing, and some basic geomorphological terms are used loosely to give a false impression of what is meant. "Erosion" is particularly used misleadingly, where measured surface lowering seems to be assumed to equal "erosion" (sensu removal of debris by a transporting medium).**

We agree that our use of certain terminology may have been confusing. We have replaced erosion by surface lowering and mass transport, depending on the meaning. Other terminology that we changed systematically includes 'lower loose moraine' to '(lower) debris apron', 'upper firm moraine' to 'upper gullied moraine' and 'tumbling rocks' to 'rockfall'.

**Second, I think there is a problem with the research design. It seems the authors have collected their high-resolution topographic data, calculated the DEM difference maps, and then attempted to interpret them on the basis of elevation change. There is no geomorphological assessment of the sites on the ground. This results in some strange and probably incorrect interpretations which even quite short field visits would have corrected. Better to observe details in the field, map the features, then use that as a basis for carrying out the topographic analysis which then quantifies processes which have already been identified and interpreted empirically, and not by inference from remotely-sensed data. My detailed comments below give examples of this problem.**

We believe this is a bit a matter of taste. We have access to this unique UAV dataset that we primarily collected to understand glacier melt. However our field observations of the moraines in combination with the spectacular dataset motivated the design of this research.

We agree that field-based geomorphological assessments of a glacier surface and adjacent moraines are essential to determine processes that drive the sourcing and transport of debris on debris-covered glaciers. For that reason we have referred to these studies in our approach (e.g. Benn & Owen, 2002; Boulton, 1978; Kirkbride & Deline, 2013). We however believe that a spatial assessment of elevation using repeat DEMs allows us to add insights to such an approach.

Furthermore, we have of course carried out field visits on site, accompanying each of the UAV flights, which have resulted in several publications (Brun et al., 2016; Miles, Steiner, & Brun, 2017; Steiner et al., 2015). We agree though that there was room for improvement and we are grateful for your

feedback as established expert in this field. We have included further explanations retrieved from the field to back up our observations made from the DEM data, e.g. at P9 (L1-3) and P10 (L18-24).

**Page/line**

**1/3. Replace "or" with "and"**

OK.

**2/3. Use the standard term for this thickness: "critical thickness". Perhaps acknowledge that this value is actually a variable, and what Østrem found in Sweden may not apply to a Nepalese glacier.**

We have changed the thickness term to critical thickness. The critical thickness is variable and depends on the local climate, however it is in the range of maximum a few centimeter.

**2/16. Surely the headwall extent is a key control as well.**

We have added this to main drivers at P2 (L18).

**2/32. Here and elsewhere, frost action is hardly mentioned as a contributor to detachment of particles from moraine slopes. Lateral moraines are typically silty and frost-susceptible, and in winter and spring ice crystal growth detaches a large amount of debris which wind and wash then remove.**

The possible importance is mentioned in the Introduction at P2 (L35). A further explanation is given in section 4.4 P9 (L23-25).

**3/5. Make it clear from the outset that all rates are rates of vertical lowering, and not of horizontal retreat, both in this study and others cited (notably Curry et al. 2006).**

This is indeed the case. Due to the mis-use of 'erosion' this was confusing throughout the paper. We now use '(vertical) surface lowering' or 'elevation change' consistently to make this clear.

**3/19. Just an observation, but this latitude/longitude location does not plot anywhere near Lirung Glacier in Google Earth. I am not saying this location is incorrect, but I wonder where the error lies?**

The coordinates seem to be the correct ones. I'm not able to reproduce the error, this might have something to do with standard coordinate types used by the software? How it shows on my computer:

[Figure]

**Generally: there are no data on the gradients of the eroding proximal faces of the lateral moraines. This is important data, because for the same rate of vertical lowering, a steeper moraine face will be releasing less debris than a gentler moraine face. A quick calculation shows that for a vertical lowering of 0.31 m/yr (p.1 l.8), a 55 degree face will retreat horizontally by 0.22 m/yr but a 65 degree face by only 0.14 m/yr. If the two faces are similar height, the gentler one will release much more debris in a year. The values of vertical lowering are not therefore very comparable between different locations without some gradient data: a conversion to horizontal retreat rates of the steep moraine faces would be useful.**

We agree that the same amount of surface lowering results in a different amount of horizontal retreat under different slope angles. However, we do not agree that a conversion to horizontal retreat rates would change the results. As our results are derived from gridded DEM's, we analyze vertical elevation changes (Figure below). The figure shows two situations with a different slope but the same magnitude of surface lowering. The total amount of debris released from both slopes is therefore the same, as their horizontal surface area is also equal. An advantage of using vertical lowering rates over horizontal retreat rates, is that they can be easily used for volumetric calculations, and incorporated in glacier flow or melt models. As we focus primarily on quantifying overall volumes of debris transport this explains the choice for our approach.

[Figure]

**5/9. "Washout" zone is introducing a new term, and I question whether this is necessary because standard geomorphological terms exist already for debris slopes below retreating faces. If the dominant process is rockfall, it's a talus. If debris flow, it's a colluvial apron or cone or simply slope. If dominant process is uncertain of complex, use a simple morphological term such as debris apron. (Also p.8 l.1 and other places where "loose part of the moraine" is used: is this the same zone?)**

The terminology is changed throughout the manuscript. The previously mentioned 'washout' zone is part of the debris apron, but has some different characteristics which are related to different depositing processes. The difference and interpretation is specified in Figure 3 and P5 (L15-18).

**6/13. "decrease", not "decreases".**

Changed

**6/19 "Most elevation change occurs in the lower loose part of the moraine. . .". The terminology and interpretation of this "lower loose part" of the moraine is where I think there are the main problems with the paper. What follows is full of confusing explanation and misleading interpretations. Having said this, all could be easily corrected with some thought and circumspection about the wider setting of the study sites. What I mean will become clear in the comments below**

**6/19 – 6/20 et seq. A key finding is that the "lower loose part of the moraine" (presumably equivalent to the "washout zone") has lowered more quickly than the upper eroding face. This lowering is variously described as "erosion" of the slope (p.7 l.24) or "slumping" (p.7 l.16). The first of these is strange given that this is a depositional slope, and it should be aggrading. The second will be discussed in due course. A further curious observation is "a patch of higher erosion. . . below the firm zone" which the authors explain as wash "plunging" off the steep slope above (see below). This whole set of interpretations is riddled with problems which I will discuss in the line-by-line points below.**

We have taken great care to clarify our terminology throughout the manuscript.

Furthermore, as noted by several reviewers, we have failed to address the presence of an ice-core within the moraine in our original manuscript. While there is no actual evidence of an ice-core, the formation process of lateral moraines suggests they are likely present and field evidence from other

debris-covered glaciers in HMA exists (e.g. (Hambrey et al., 2009)). However at least for the case of Lirung and other debris-covered tongues in the Langtang catchment, field observations suggests that these ice cores are covered in a very thick mantle of debris, very likely larger than 2 m, contrary to more thinly covered moraines elsewhere (e.g. (Lukas, Nicholson, Ross, & Humlum, 2005)). During many walks in multiple field seasons on the glacier tongue, including the flanks, nowhere along or within 50 m of the foot of the moraine have we observed any ice, neither as ice cliffs nor as ice covered under thin moraine material. Furthermore, none of the debris thickness measurements ever taken in the field close to the moraine on the glacier surface where thinner than 50 cm (see e.g. (Ragettli et al., 2015)), and since the debris progressively thickens (see e.g. (Nicholson, McCarthy, Pritchard, & Willis, 2018) towards the moraine it is likely much thicker there (Nicholson et al., 2018). We estimate the maximum downwasting rate under the moraine by quantifying the elevation change in a relative flat zone of 20 meter wide close to the moraine. We removed sections which showed clear depositional features, to limit the noise caused by debris deposition. The maximum downwasting rate is approximately 0.6 m $yr^{-1}$. The top melt of buried ice declines exponentially with increasing debris-cover thickness (Östrem, 1959; Schomacker, 2008), but as debris thickness on the glacier is generally > 50 cm (McCarthy, Pritchard, Willis, & King, 2017; Ragettli et al., 2015), the decline of melt rates underneath the moraine due to the additional debris thickness is relatively small.

Assuming a maximum downwasting rate of 0.6 m $yr^{-1}$ due to the ice core, the remaining elevation change due to mass transport in this part is +0.19 m $yr^{-1}$. This results in a significantly reduced rate of material reaching the glacier surface and hence our interpretations of the results. As a consequence, we removed the explanation on page 10 (line 20-27) as suggested.

While we acknowledge that further detailed analysis could be carried out to ascertain presence of a potential ice core (i.e. GPR) or to understand the processes in recent decades and centuries, our aim here was to determine the volume of debris that moved onto the glacier surface in recent years using an UAV. We believe that the use of high-resolution DEMs in combination with geomorphological analysis has great potential to understand the dynamics of debris-covered glacier tongues.

**6/20-22. I find this comparison of lowering rates confusing. It needs to be more clearly stated what is being compared with what.**

The comparison is indeed confusing, as the rates found by Watanabe et al. (1998) were reported from a different field situation including the valley rockwall. We have therefore removed this comparison from the manuscript.

**6/23-24. These differences could be the result of slope angle differences (see above).**

See response to concern above.

**6/28. Delete "m yr-1" after "0.02".**

Deleted

**7/3. "slumps and rockfall" ?**

Changed. The word 'or' indicates an indistinguishable process, which is certainly not the case here.

**7/6 – 7. If I am correct in what this sentence is referring to (and I'm uncertain about this after looking at Figure 5A), I wonder whether this refers to a short steep step of fresh-looking till which commonly forms the very base of the eroding till cliff, and right at the very top of the debris apron below. This is a very common feature of lateral moraines above thinning debris-covered ice generally. Though I haven't read about it in the literature, it has always seemed to me that it forms by lowering of the debriscovered ice at the base of the steep moraine wall: nothing more complicated than that. This paper actually seems to demonstrate this, because the debris aprons are lowering and not aggrading upwards. This just creates a short slope segment marking (probably) that melt season's lowering and separation of the debris slope below from the steep moraine face (the "firm zone"?) above. It's nothing to do with greater erosion by water plunging off the wall above: it's too continuous and uniform to be caused by this. All the evidence presented strongly suggests that the debris apron isn't eroding, but being gently let down by ablation of glacier ice underneath. Whether this ice is still connected to the glacier or not is secondary: the important point is that its surface is lowering. (This, after all, is what exposes the proximal moraine faces in the first place).**

We added this suggested interpretation to the manuscript at P7 (L18-21). This process is certainly a possible explanation, but the area of increased surface lowering continues further down the debris apron and changes gradually, which does not stroke with the formation of a steep 'step'. Also, there seems to be more surface lowering between more intensely gullied sections, and therefore we cannot exclude the possibility that these higher rates are (also) related to water flow processes.

**7/13. Replace "tumbling rocks" with "rock fall". Oversteepening of the slope isn't by water flow processes: it is by downward extension of the slope base as the ice below the slope thins.**

Done and it has been included.

**7/14-15. "The slump toe. . ..the event": meaning unclear. What slump? What do you mean by "the moraine"? After what "event? Confusing. "The lower part of the moraine. . ..": are you including the debris apron ("washout zone/ lower loose part") as part of the moraine? If so, why? It isn't part of the moraine once the debris has been removed from the face and redeposited below.**

The slump is visible in Figure 6, which is now specified. The terminology here is changed as well, which now indicates that the slump deposit can be found on the coalescing debris below the upper moraine.

**7/17-18. Average velocities of what? What process is faster than creep? Creep of what? Why make this comparison? Confusing.**

The debris on the talus has a lateral displacement towards the glacier. This rate is compared to the rate of several processes to indicate those processes that are related to the displacement. We have clarified that in the paper on P7-8 (L31, L1-6).

**7/20. Permafrost requires seasonally-frozen ground: it doesn't require permafrost. On p8 l.2 you show it's probably not solifluction anyway, because motion is greatest in the wet summer season, so it's slow slump or creep of unfrozen saturated debris.**

Thank you for the correction by email afterwards! We changed the argumentation of this part and now explicitly exclude solifluction from the likely processes. We also argue that creep of unfrozen debris is

an unlikely option, as the displacement rates related to creep generally are lower than those observed here. Changes were made in P7-8 (L31, L1-6).

**7/24. "erosion rates" of the "loose part of the moraine"? Is this not surface lowering due to a melting ice core, with slow downslope movement of the debris cover over the ice beneath? In other words, it's behaving like a wasting ice-cored moraine. It cannot be called "erosion" because no debris is being entrained and removed by an external medium. Use of this term is misleading and confusing. It's slow gravitational transport.**

The term erosion was often incorrectly used for elevation change rates. This is improved throughout the manuscript as indicated above.

**8/6. Freeze-thaw cycles are important for causing needle-ice growth and detachment of particles in moraine faces. A problem is that few scientists observe moraine faces in winter and spring, when ice crystal growth is evident and thaw-saturation of silty till gives it a very different consistency to the indurated, dry material we see in dry summer weather.**

While we don't have precise measurements of any such process we definitely observed it in the field on shaded lateral moraines and the difference in consistency is apparent while probing the material. While we can acknowledge that qualitatively, it is difficult to account for that in actual estimates of debris transport. This is specified in P9 (L23-25).

**8/9-12. Again your "erosion" values are slope-angle dependent. They are a lowering value, not erosion rates. As discussed above, the lower slope isn't actually eroding at all, so this comparison is spurious.**

As discussed above, we do not think that the values presented are slope dependent. The term erosion is changed to surface lowering/elevation change. See our earlier comments.

**8/16. Is the "lower moraine" the same as the washout zone, loose part, etc? There needs to be clarity and consistency of terminology throughout the paper. A schematic diagram showing a cross-section through the moraine and toe area with the different zones labelled would be helpful.**

As pointed out above, we have now revised our terminology throughout the paper. A diagram is added to Figure 3.

**8/16-17. Again, it is being assumed that lowering of the lower gentler depositional slope is by erosion, but no geomorphological evidence of erosional process. The exception is when glacier confluences form medial moraines from lateral moraines in-transport (sensu Boulton 1978), which then spread debris across the centre of the ablation zone by secondary dispersal (sensu Kirkbride & Deline 2013) to form full-width debris covers. By this mechanism complete debris covers can form. Perhaps this is worth a mention in the wider implications: but the debris needs to be introduced to the glacier centre some distance upstream.**

We now acknowledge the likeliness of ice beneath the debris apron throughout the manuscript. Morphological evidence of debris flow and water flow erosion is presented in section 4.4, which would certainly be able to entrain, rework and remove sediment from the debris apron below. Therefore we do not completely remove the erosional processes on the debris apron from the manuscript. The

possibility of forming a complete debris cover of lateral moraine derived debris by medial moraine formation is added in line P10 (L20-23).

**9/15 "approximately"**

Changed

**Section 4.6 Clast Analysis. This is all fine, but how did you assess the roundness of rounded moraine clasts which have shattered on impact when falling from the moraine, to give some angular edges? Did you measure the sharpest edge or the most rounded?**

Thank you for pointing this out, this was indeed a question raised during the analysis of the clasts in the field. When investigating the clasts, we looked at the rounded edges, hence clasts that were round but broken into smaller parts due to possible impact were classified as round. Unfortunately we did not however record the number of clasts that were possibly shattered upon impact, also because we found it hard to distinguish which clasts were shattered locally (frost or impact of larger boulders moving locally?) and which ones had fallen from higher up the moraine. For a more detailed analysis this may however be an issue to look into also on Lirung.

**10/11-14. This calculation should be omitted because lateral moraine supply is clearly demonstrated not to cover the full glacier width, so it is a pointless calculation. Also, your average rate of 0.31 m yr-1 isn't valid because it includes the debris apron below the moraine (assuming I have read p.6 l.16-20 correctly), which I have argued earlier is a depositional slope whose lowering is not due to erosion.**

We included the effect of ice melt lowering on the debris apron. Assuming a lower melt rate underneath the thick debris apron (Schomacker, 2008), an approximation of the maximum debris supply is currently made using only erosion from the upper gullied slopes. This results in a debris thickness increase of 5 cm $yr^{-1}$ instead of 17 cm $yr^{-1}$. The calculation including the maximum modelled runout length is kept to give an indication for other glaciers, where runout may be greater. However, we do acknowledge in the text that this is not the case for our study area.

**10/14. An annual debris thickness increase of 0.29 m yr-1 would generate a layer 10 m thick in about 35 years. So do you see debris layers this thick over the margins of the glacier? I very much doubt it. This reinforces that your rate is not based on a valid calculation. Yet (l.16) you persist in arguing that it is correct.**

As indicated above, the rate is reduced to 0.05 m $yr^{-1}$, which is more in line with other supply rates and debris thicknesses found.

**10/20. Point (b) is simply not a realistic process. Debris accumulating below most of the lateral moraines is not affected by terminus behaviour.**

We agree that the process mentioned under (b) is irrelevant for most of the glacier, which is indeed not affected by processes at the glacier terminus. This section is removed from the manuscript.

**10/27. Point (c) ditto. This whole set of justifications is spurious.**

Point (c) is rewritten, and now suggests that the form of the glacier terminus might also be influenced by the lateral moraine supply.

**11/1-6. Overstates the significance of lateral moraine near a terminus. To effectively create a complete debris cover which will affect glacier mass balance, debris needs to be introduced to the transport system much higher upstream. Debris introduced from lateral sources close to the terminus is of more geomorphological than glaciological interest, by creating ice-cored landforms along the base of lateral moraines (which are increasingly common as glaciers retreat). It is most significant in the later stages of the decay of glacier tongues, so a comparison with models of active debris-covered glaciers (p.11 l.4) is misleading. Overall, my view (reluctantly) is that the paper needs a very careful rethink and rewrite to possibly then be acceptable for publication.**

We agree that moraine development and processes are of geomorphological interest, however we believe that in our attempt to understand debris-covered tongues better they are also of interest from a glaciological perspective. It is certainly true that initial source of debris are the headwalls or the bedrock. We hope we have made this clearer now. We acknowledge that the moraine material is just remobilized material. However, we observe constant transport of debris down lateral moraines in our field site. In addition thickening rates at the terminus of debris-covered tongues seem to be faster than what englacial transport could provide (Gibson et al., 2017). Therefor we hypothesized that remobilized material from the moraines also plays a role, even near the terminus. As we show, it does so primarily at the margins, and this may be a reason why ice-cliffs and lakes are primarily found towards the glacier center line.

**References**

Benn, D. I., & Owen, L. A. (2002). Himalayan glacial sedimentary environments: A framework for reconstructing and dating the former extent of glaciers in high mountains. *Quaternary International*, *97–98*, 3–25. https://doi.org/10.1016/S1040-6182(02)00048-4

Boulton, G. S. (1978). Boulder shapes and grain-size distributions of debris as indicators. *Sedimentology*, *25*, 773–799.

Brun, F., Buri, P., Miles, E. S., Wagnon, P., Steiner, J., Berthier, E., … Pellicciotti, F. (2016). Quantifying volume loss from ice cliffs on debris-covered glaciers using high-resolution terrestrial and aerial photogrammetry. *Journal of Glaciology*, *62*(234), 684–695. https://doi.org/10.1017/jog.2016.54

Gibson, M. J., Glasser, N. F., Quincey, D. J., Mayer, C., Rowan, A. V., & Irvine-Fynn, T. D. L. (2017). Temporal variations in supraglacial debris distribution on Baltoro Glacier, Karakoram between 2001 and 2012. *Geomorphology*, *295*, 572–585. https://doi.org/10.1016/j.geomorph.2017.08.012

Hambrey, M. J., Quincey, D. J., Glasser, N. F., Reynolds, J. M., Richardson, S. J., & Clemmens, S. (2009, December). Sedimentological, geomorphological and dynamic context of debris-mantled glaciers, Mount Everest (Sagarmatha) region, Nepal. *Quaternary Science Reviews*. Elsevier Ltd. https://doi.org/10.1016/j.quascirev.2009.04.009

Kirkbride, M. P., & Deline, P. (2013). The formation of supraglacial debris covers by primary dispersal from transverse englacial debris bands. *Earth Surface Processes and Landforms*, *38*(15), 1779–1792. https://doi.org/10.1002/esp.3416

Lukas, S., Nicholson, L. I., Ross, F. H., & Humlum, O. (2005). Formation, meltout processes and

landscape alteration of high-arctic ice-cored moraines - Examples from Nordenskiöld land, central Spitsbergen. *Polar Geography*, *29*(3), 157–187. https://doi.org/10.1080/789610198

McCarthy, M., Pritchard, H., Willis, I., & King, E. (2017). Ground-penetrating radar measurements of debris thickness on Lirung Glacier, Nepal. *Journal of Glaciology*, *63*(239), 543–555. https://doi.org/10.1017/jog.2017.18

Miles, E. S., Steiner, J. F., & Brun, F. (2017). Highly variable aerodynamic roughness length (z0) for a hummocky debris-covered glacier. *Journal of Geophysical Research: Atmospheres*, *122*(16), 8447–8466. https://doi.org/10.1002/2017JD026510

Nicholson, L. I., McCarthy, M., Pritchard, H., & Willis, I. (2018). Supraglacial debris thickness variability: Impact on ablation and relation to terrain properties. *The Cryosphere Discussions*, *12*(12), 1–30. https://doi.org/10.5194/tc-2018-83

Östrem, G. (1959). Ice Melting under a Thin Layer of Moraine, and the Existence of Ice Cores in Moraine Ridges. *Geografiska Annaler*, *41*(4), 228–230. https://doi.org/10.1080/20014422.1959.11907953

Ragettli, S., Pellicciotti, F., Immerzeel, W. W., Miles, E. S., Petersen, L., Heynen, M., … Shrestha, A. (2015). Unraveling the hydrology of a Himalayan catchment through integration of high resolution in situ data and remote sensing with an advanced simulation model. *Advances in Water Resources*, *78*, 94–111. https://doi.org/10.1016/j.advwatres.2015.01.013

Schomacker, A. (2008). What controls dead-ice melting under different climate conditions? A discussion. *Earth-Science Reviews*, *90*(3–4), 103–113. https://doi.org/10.1016/j.earscirev.2008.08.003

Steiner, J. F., Pellicciotti, F., Buri, P., Miles, E. S., Immerzeel, W. W., & Reid, T. D. (2015). Modelling ice-cliff backwasting on a debris-covered glacier in the Nepalese Himalaya. *Journal of Glaciology*, *61*(229), 889–907. https://doi.org/10.3189/2015JoG14J194

---

## Author Comment (AC6) · 30 Jan 2019

**Response to RC4**

Dear reviewer,

Thank you for your constructive comments, which we address in detail below. First, we respond to the general comments, after which the detailed comments are addressed point by point.

**General comments: The hypothesis stated on page 3 of the manuscript proposes that lateral moraines can play an important role in supplying the glacier tongue with debris, especially for glaciers that are stagnating and the tongue is disconnected from the headwall. The goals of the paper are to quantify erosion rates across a 5-year period by using high-resolution DEMs. The dataset provided by this paper is remarkable, very high resolution and from a remote location whose future demise will wreak havoc on downstream communities. This large, alpine glacier undoubtedly warrants current research and the negative feedback cycle created by debris insulating the glacier tongue is an understudied process.**

**Overall, I find that the study is both overly complex and overly simplified at the same time. To improve the manuscript, the authors could remove some datasets, such as the clast shape analysis, as the population of clasts sampled seems far too small to gain any interpretable insights. This would remove the field-based component of the manuscript, but I don't think that would weaken the paper. On the contrary, I think the paper could be strengthened by using only the remotely sensed data.**

We appreciate your concern that additional data complicates our work but we strongly believe that ground validation of observations made from the UAV data is necessary and this was raised as one of the key points of reviewer 3. We examined over 3000 clasts, which is more than the site specific sample size used in Benn and Ballantyne (1994) & Lukas et al. (2013), which proposed the methods used in this paper. Therefor we assume that our sample size is large enough to draw conclusions from, and we do believe this data strengthens our message.

**The authors present many erosion rate calculations for different sections of the moraine, but they never use those data to predict how long (in years, e.g.) it would take for the glacier tongue to remove a certain amount of debris. Although not part of the manuscript in its current state, I think the manuscript could be improved it the authors added a predictive component to their research, like answers questions such as: if the debris cover is adding a negative feedback to the glacier ice, will the debris allow the ice to persist past the year that the glacier ice is expected to melt? And if so, by how many years? At what time point would the lateral moraines deflate enough so that they are not contributing debris to the glacier tongue?**

Although certainly interesting questions, indicating such precise relations between glacial melt and supraglacial debris cover requires extensive knowledge of other sources of debris, the distribution of lateral moraine derived debris over the glacier, and finally the feedbacks between debris cover and glacial melt. The knowledge and data required to do this accurately is unfortunately unavailable at present and beyond the scope of the our study.

With the approximation of lateral moraine sediment input to the glacier we want to contribute to the current discussion, with a first indication of the spatial and temporal variability lateral moraine sediment supply to a glacier tongue in High Mountain Asia. This will hopefully allow us in the future to include these estimates into projections of the development of such glaciers.

**Apart from the technical issues I present below, I find that the authors use the term 'erosion' in a misleading way, and could replace it with 'transport' in some situations. Erosion implies that the material is carried away from the glacier system, as opposed to being transported to lower on the glacier tongue, thus leading to the well-insulated glacier snout that they discuss. If the authors have a mechanism to prove that the material is eroded than they can continue using that term, otherwise it could be replaced with transport.**

We agree with the reviewer that the term erosion is not used correctly throughout the paper, and this issue was also flagged by reviewer 3. We have used it interchangeably for changes in elevation, transport process and erosion (entrainment and removal of material by an external medium). We adapted the terminology throughout the manuscript. Elevation differences are now referred to as 'surface lowering' or 'elevation change' and processes such as slumping are referred to as 'mass transport' or 'transport processes'. At some locations, for example in the gullied upper part, actual erosion takes place and we continue to use the term.

**Specific comments:**

**As for the writing of the manuscript, I think lines 1-16 on p. 3 can be written so that a testable hypothesis is proposed. The authors list approximate erosion rates from the Alps and from Norway, which could be higher due to the contribution of headwall erosion. The authors need to articulate what they are testing, instead of stating their goals. Perhaps they could add "by quantifying erosion rates of lateral moraines from a DCG in Nepal, we will test the hypothesis that glaciers with disconnected tongues will have lower erosion rates than those with tongues connected to the headwalls". Lines 12-16 on page 3 do a good job of explaining the goals of the paper, but it is lacking a testable hypothesis.**

The inclusion of a hypothesis is a good suggestion. Since the comparison of the erosion rates has changed as result of comments of other reviewers we have defined one hypothesis related the importance of debris supply from lateral moraines on debris covered glaciers in lines P3 (L15-17).

**Technical corrections:**

**Page 2, Line 31: The phrase "gullied upper part" is vague, I would suggest changing it to more descriptive language. I don't get a sense of what part of the moraines are gullied.**

The term 'gully' or 'gullied' is generally used in moraine geomorphology for the distinct features of the upper part of moraines, see for example (Draebing and Eichel 2018; Lukas et al. 2012). This was not very well visible in the otherwise too small Figure 3, which has now been enlarged to make this clear. We have added this in the respective sentence.

**Page 2, Lines 34-35: The sentence: "The transport processes on the moraines are most active directly after deglaciation" makes intuitive sense but is a large claim. Does this imply that your modern study falls within the period of most active transport? I would either add another sentence or two linking this claim to your study, or remove it all together.**

We decided to remove the sentence, as we do not further elaborate on the stage the moraine is in.

**Page 4 line 10, that is a big assumption that no melting core exists in the moraine. Do they have any data that would substantiate this claim?**

As noted by several reviewers, we have failed to address the presence of an ice-core within the moraine in our original manuscript. While there is no actual evidence of an ice-core, the formation process of lateral moraines suggests they are likely present and field evidence from other debriscovered glaciers in HMA exists (e.g. (Hambrey et al. 2009)). However at least for the case of Lirung and other debris-covered tongues in the Langtang catchment, field observations suggests that these ice cores are covered in a very thick mantle of debris, very likely larger than 2 m, contrary to more thinly covered moraines elsewhere (e.g. (Lukas et al. 2005)). During many walks in multiple field seasons on the glacier tongue, including the flanks, nowhere along or within 50 m of the foot of the moraine have we observed any ice, neither as ice cliffs nor as ice covered under thin moraine material. Furthermore, none of the debris thickness measurements ever taken in the field close to the moraine on the glacier surface where thinner than 50 cm (see e.g. (Ragettli et al. 2015)), and since the debris progressively thickens (see e.g. (Nicholson et al. 2018) towards the moraine it is likely much thicker there (Nicholson et al. 2018). We estimate the maximum downwasting rate under the moraine by quantifying the elevation change in a relative flat zone of 20 meter wide close to the moraine. We removed sections which showed clear depositional features, to limit the noise caused by debris deposition. The maximum downwasting rate is approximately 0.6 m yr$^{-1}$. The top melt of buried ice declines exponentially with increasing debris-cover thickness (Östrem 1959; Schomacker 2008), but as debris thickness on the glacier is generally > 50 cm (Ragettli et al. 2015; McCarthy et al. 2017), the decline of melt rates underneath the moraine due to the additional debris thickness is relatively small.

Assuming a maximum downwasting rate of 0.6 m yr$^{-1}$ due to the ice core, the remaining elevation change due to mass transport in this part is +0.19 m yr$^{-1}$. This results in a significantly reduced rate of material reaching the glacier surface and hence our interpretations of the results. As a consequence, we removed the explanation on page 10 (line 20-27) as suggested.

While we acknowledge that further detailed analysis could be carried out to ascertain presence of a potential ice core (i.e. GPR) or to understand the processes in recent decades and centuries, our aim here was to determine the volume of debris that moved onto the glacier surface in recent years using an UAV. We believe that the use of high-resolution DEMs in combination with geomorphological analysis has great potential to understand the dynamics of debris-covered glacier tongues.

**Page 6, Lines 12-15: This sentence reads as a hypothesis since the authors are predicting what the . It reads as more of a topic sentence and could be moved to the beginning of the paragraph.**

This section is rewritten to first mention the goals and hypothesis, and afterwards the methods used to test this.

**Page 6, Lines 19-20: Here the terms "lower loose" and "upper firm" start getting usage as descriptors for parts of the moraine. I understand that terms loose and firm correlate to the different erosion rates, but perhaps more descriptive terms could be used, such as "low erosion area" and "higher erosion area".**

The terminology of these parts is changed throughout the manuscript, and is more descriptive in terms of the dominant active processes and geomorphological type. The upper firm part is referred to as upper gullied area, the lower part is currently a debris apron or a slope with coalescing debris cones. The terms used in the paper are indicated in Figure 3.

**Page 7, Lines 19-20: Solifluction is mentioned as a main transport mechanism, yet little discussion in the text is allocated to this process. Is there a way the authors can interpret solifluction processes from their DEM data?**

As indeed the signs for solifluction were unclear, and the speedup of horizontal displacement in summer months does not support solifluction as the main transport mechanism, we only refer to the transport mechanism of 'slow slumping' in this regard and have adapted this in the manuscript.

**Page 11, Line 13: Change mechanisms to mechanism**

Changed

**Page 11, Line 22: Change are to is**

Changed

**Figures 4 and 5: It would be useful to have an index map of where these images/data are located on the glacier.**

The location of these insets is added to Figure 2.

**References**

Benn, Douglas I., and Colin K. Ballantyne. 1994. "Reconstructing the Transport History of Glacigenic Sediments: A New Approach Based on the Co-Variance of Clast Form Indices." *Sedimentary Geology* 91 (1–4): 215–27. https://doi.org/10.1016/0037-0738(94)90130-9.

Draebing, Daniel, and Jana Eichel. 2018. "Divergence, Convergence, and Path Dependency of Paraglacial Adjustment of Alpine Lateral Moraine Slopes." *Land Degradation and Development* 29 (6): 1979–90. https://doi.org/10.1002/ldr.2983.

Hambrey, Michael J., Duncan J. Quincey, Neil F. Glasser, John M. Reynolds, Shaun J. Richardson, and Samuel Clemmens. 2009. "Sedimentological, Geomorphological and Dynamic Context of Debris-Mantled Glaciers, Mount Everest (Sagarmatha) Region, Nepal." *Quaternary Science Reviews*. Elsevier Ltd. https://doi.org/10.1016/j.quascirev.2009.04.009.

Lukas, Sven, Douglas I. Benn, Clare M. Boston, Martin Brook, Sandro Coray, David J.A. Evans, Andreas Graf, et al. 2013. "Clast Shape Analysis and Clast Transport Paths in Glacial Environments: A Critical Review of Methods and the Role of Lithology." *Earth-Science Reviews*. Elsevier. https://doi.org/10.1016/j.earscirev.2013.02.005.

Lukas, Sven, Andreas Graf, Sandro Coray, and Christian Schlüchter. 2012. "Genesis, Stability and Preservation Potential of Large Lateral Moraines of Alpine Valley Glaciers - towards a Unifying Theory Based on Findelengletscher, Switzerland." *Quaternary Science Reviews* 38: 27–48. https://doi.org/10.1016/j.quascirev.2012.01.022.

Lukas, Sven, Lindsey I. Nicholson, Fionna H. Ross, and Ole Humlum. 2005. "Formation, Meltout Processes and Landscape Alteration of High-Arctic Ice-Cored Moraines - Examples from Nordenskiöld Land, Central Spitsbergen." *Polar Geography* 29 (3): 157–87. https://doi.org/10.1080/789610198.

McCarthy, Michael, Hamish Pritchard, Ian Willis, and Edward King. 2017. "Ground-Penetrating Radar Measurements of Debris Thickness on Lirung Glacier, Nepal." *Journal of Glaciology* 63 (239): 543–55. https://doi.org/10.1017/jog.2017.18.

Nicholson, Lindsey I., Michael McCarthy, Hamish Pritchard, and Ian Willis. 2018. "Supraglacial Debris Thickness Variability: Impact on Ablation and Relation to Terrain Properties." *The Cryosphere Discussions* 12 (12): 1–30. https://doi.org/10.5194/tc-2018-83.

Östrem, Gunnar. 1959. "Ice Melting under a Thin Layer of Moraine, and the Existence of Ice Cores in Moraine Ridges." *Geografiska Annaler* 41 (4): 228–30. https://doi.org/10.1080/20014422.1959.11907953.

Ragettli, S., F. Pellicciotti, W. W. Immerzeel, E. S. Miles, L. Petersen, M. Heynen, J. M. Shea, D. Stumm, S. Joshi, and A. Shrestha. 2015. "Unraveling the Hydrology of a Himalayan Catchment

through Integration of High Resolution in Situ Data and Remote Sensing with an Advanced Simulation Model." *Advances in Water Resources* 78 (April): 94–111. https://doi.org/10.1016/j.advwatres.2015.01.013.

Schomacker, Anders. 2008. "What Controls Dead-Ice Melting under Different Climate Conditions? A Discussion." *Earth-Science Reviews* 90 (3–4): 103–13. https://doi.org/10.1016/j.earscirev.2008.08.003.

---

## Author Response (AR2)

Dear Niels,

Thank you taking care of the editorial process these last weeks, despite previous problems. We appreciate your efforts to complete and improve the manuscript. Below you find our reply to your comments.

**Change title to: Sediment supply from lateral moraines to a debris covered glacier in the Himalaya.**
Changed.

**Drop the word 'tongues' from the abstract. Simply use 'glaciers'.**
We have changed this throughout and adapted some lines to avoid unnecessary repetions.

**Mention structure-from-motion as the technique used to construct DEMs. This is a key search term for many colleagues.**
Thank you for the suggestion. We have added this.

**Line 8: make clear that the elevation change reported here is within the moraines.**
Changed.

**Line 10: When reporting seasonal rates, then a time unit smaller than yr is appropriate. Consider using month instead. This is not an instruction. Feel free to decide otherwise.**
We do agree that a time unit smaller than yr is better in indicating the rates are seasonal, or half yearly. However, monthly rates suggest we have data at a monthly resolution, which is not the case. Furthermore, in other glacier related erosion processes as rockwall or basal erosion values are expressed on a yearly timescale, and the direct comparison essential in our Discussion is easier without deviating from this norm.

**Line 12: can you give a range on the debris thickness?**
We added a value for the absolute thickness in the abstract, which is based on a number of point observations. We specifically refer to the observations with more detail also at p11/l23.

**Line 12: ... possibly reducing melt rates of underlying glacier ice.**
Added.

**The legends of the picture boxes in Figs. 2 and 3 are not clearly legible, due to lack of contrast. Please use a background that provides more contrast.**
Changed.

**Figure 9C needs more space, and/or larger symbols and lettering.**
Figure, font size and dot size enlarged.

With kind regards,

Teun